# The use of precision diagnostics for monogenic diabetes: a systematic review and expert opinion

Rinki Murphy [1,2,208✉], Kevin Colclough[3,208], Toni I. Pollin[4,208], Jennifer M. Ikle[5,6], Pernille Svalastoga[7,8], Kristin A. Maloney [4], Cécile Saint-Martin[9], Janne Molnes[8,10], ADA/EASD PMDI*, Shivani Misra[11,12], Ingvild Aukrust[8,10], Elisa de Franco[13], Sarah E. Flanagan [13], Pål R. Njølstad [7,8], Liana K. Billings [14,15,209], Katharine R. Owen[16,17,209] & Anna L. Gloyn [5,6,18,209✉]

## Abstract

**Background** Monogenic diabetes presents opportunities for precision medicine but is underdiagnosed. This review systematically assessed the evidence for (1) clinical criteria and (2) methods for genetic testing for monogenic diabetes, summarized resources for (3) considering a gene or (4) variant as causal for monogenic diabetes, provided expert recommendations for (5) reporting of results; and reviewed (6) next steps after monogenic diabetes diagnosis and (7) challenges in precision medicine field.

**Methods** Pubmed and Embase databases were searched (1990-2022) using inclusion/exclusion criteria for studies that sequenced one or more monogenic diabetes genes in at least 100 probands (Question 1), evaluated a non-obsolete genetic testing method to diagnose monogenic diabetes (Question 2). The risk of bias was assessed using the revised QUADAS-2 tool. Existing guidelines were summarized for questions 3-5, and review of studies for questions 6-7, supplemented by expert recommendations. Results were summarized in tables and informed recommendations for clinical practice.

**Results** There are 100, 32, 36, and 14 studies included for questions 1, 2, 6, and 7 respectively. On this basis, four recommendations for who to test and five on how to test for monogenic diabetes are provided. Existing guidelines for variant curation and gene-disease validity curation are summarized. Reporting by gene names is recommended as an alternative to the term MODY. Key steps after making a genetic diagnosis and major gaps in our current knowledge are highlighted.

**Conclusions** We provide a synthesis of current evidence and expert opinion on how to use precision diagnostics to identify individuals with monogenic diabetes.

## Plain Language Summary

Some diabetes types, called monogenic diabetes, are caused by changes in a single gene. It is important to know who has this kind of diabetes because treatment can differ from that of other types of diabetes. Some treatments also work better than others for specific types, and some people can for example change from insulin injections to tablets. In addition, relatives can be offered a test to see if they are at risk. Genetic testing is needed to diagnose monogenic diabetes but is expensive, so it's not possible to test every person with diabetes for it. We evaluated published research on who should be tested and what test to use. Based on this, we provide recommendations for doctors and health care providers on how to implement genetic testing for monogenic diabetes.

A full list of author affiliations appears at the end of the paper.

The use of precision diabetes medicine has gained increased awareness to improve diagnosis and treatment for patients with diabetes[1]. While the majority of those living with diabetes globally have polygenic disorders categorized as type 1 diabetes (the predominant form in those diagnosed in childhood and early adulthood), or type 2 diabetes (the predominant form in older people), approximately 1-2% have monogenic forms of diabetes, which is most commonly found in diabetes arising in neonates through to young adulthood[2]. Knowledge of the exact molecular defect and mechanism of disease is crucial for precision diagnostics, which informs treatment, prognostics, and monitoring. Identification of monogenic diabetes, of which there are now over 40 different genetic subtypes, have led to improved insight into the mechanism of disease and enabled precision diabetes treatment for several of these disorders, e.g., sulfonylurea agents for the treatment of $K_{ATP}$ neonatal diabetes[3,4], HNF1A-diabetes and HNF4A-diabetes[5–7]. The genetic diagnosis of any given monogenic diabetes subtype informs precision prognostics e.g., lack of microvascular or macrovascular complications in those with a heterozygous GCK etiology and informs precision monitoring particularly in syndromic forms where the genetic diagnosis precedes the development of additional clinical features such as hepatic dysfunction and skeletal dysplasia in *EIF2AK3* or hearing and vision loss in *WFS1*[8,9]. Thus, diagnosing monogenic diabetes presents an opportunity to identify those who would benefit from precision medicine.

There are, however, key knowledge gaps that are obstacles for precision diagnostics in monogenic diabetes. The clinical diagnosis of diabetes is based on the measurement of a single molecule, glucose. The correct classification of diabetes relies on differentiating based on overlapping clinical features such as age, body mass index (BMI), history of diabetic ketoacidosis, glycemic response to non-insulin therapies and the selective use of C-peptide and autoantibodies[10]. These features are less reliable for correct diabetes classification in people of non-European ancestry, in whom the prevalence of type 2 diabetes is usually greater and often occurs from a younger age than in Europeans. The term maturity-onset diabetes of the young (MODY), frequently used to refer to common monogenic diabetes has three classical criteria: autosomal dominant inheritance pattern, onset of diabetes before 25 years, and non-insulin dependence (due to residual beta cell function)[11]. However, these are not specific as they overlap with the clinical features seen in type 1 and type 2 diabetes[12]. These classical criteria are also not sensitive, since there are spontaneous mutations occurring in individuals without a family history of diabetes, autosomal recessive cases[13–15], later onset cases of monogenic diabetes and frequent requirement for insulin treatment. The term MODY originates from the time when the terms juvenile-onset and maturity-onset were used to distinguish between type 1 and type 2 diabetes and does not precisely distinguish the various phenotypes associated with the numerous genetic etiologies for monogenic diabetes subtypes[16]. Recent studies show that people with monogenic diabetes are often misdiagnosed as having type 1 diabetes or type 2 diabetes[17]. Given the currently prohibitive cost and low yield of universal genetic testing in the vast majority with clinically classified type 1 and type 2 diabetes[12,18–20], there is therefore a need for more knowledge on who to test for monogenic diabetes using various clinical and biomarker-based criteria that increase the yield for this diagnosis, thereby, making such genetic testing more cost-effective.

Recent breakthroughs in sequencing technologies make it possible to sequence the entire genome of a person in less than a day[21,22]. Routine genome sequencing may not be appropriate for diagnosing monogenic diabetes due to costs, interpretation challenges, and ethical issues in reporting incidental findings[23].

Less resource-demanding technologies are exome sequencing, panel exome sequencing and next-generation sequencing (NGS) using a targeted panel where many or all monogenic diabetes genes can be investigated simultaneously[24]. In some instances, e.g. diagnosing a known disease-causing variant in additional family members, traditional Sanger sequencing might be preferred due to economy, speed, and reliability. The use of real-time PCR such as for detecting and quantifying mitochondrial m.3243A>G variant load, droplet digital PCR for analysis of both paternally and maternally inherited fetal alleles, copy number variant analysis for detecting gene deletions and methylation sensitive assays (e.g., for 6q24 abnormalities as a common cause of transient neonatal diabetes) are all available technologies. Thus, there are knowledge gaps regarding the choice of technology being a balance between cost, time, the degree of technical, scientific and bioinformatic expertise required, and the performance/diagnostic yield in particular diagnostic settings.

Best practices have been developed on how to report genetic findings[25]. The results of genetic tests may, however, be challenging to interpret[26]. Identifying a pathogenic variant may confirm a diagnosis of monogenic diabetes, indicate that a person is a carrier of a particular genetic variant, or identify an increased risk of developing diabetes. Although a "no pathogenic variant identified" test result often excludes a common monogenic etiology, it is quite possible for a person who lacks a known pathogenic variant to have or be at risk for alternative monogenic types of diabetes–sometimes because of limitations in technology but often due to inability to anticipate all possible genes that might be involved and limitations in our ability to interpret them depending on the technology used. In some cases, a test result might not give any useful information being uninformative, indeterminate, or inconclusive. If a genetic test finds a variant of unknown significance (VUS), it means there is not enough scientific information to confirm or refute causality of monogenic diabetes, or data are conflicting[27]. Two expert panels have formed to develop guidelines for reviewing evidence to determine which genes (ClinGen Monogenic Diabetes Gene Curation Expert Panel [MDEP GCEP, https://clingen.info/affiliation/40016/]) and gene variants (MDEP VCEP, https://clinicalgenome.org/affiliation/50016/) are considered causative of monogenic diabetes. However, the implementation of these guidelines by the many diagnostics laboratories around the world is likely to be variable.

The Precision Medicine in Diabetes Initiative (PMDI) was established in 2018 by the American Diabetes Association (ADA) in partnership with the European Association for the Study of Diabetes (EASD)[28]. The ADA/EASD PMDI includes global thought leaders in precision diabetes medicine who are working to address the burgeoning need for better diabetes prevention and care through precision medicine[29]. This systematic review is written on behalf of the ADA/EASD PMDI as part of a comprehensive evidence evaluation in support of the 2nd International Consensus Report on Precision Diabetes Medicine[30].

To investigate the evidence for who to test for monogenic diabetes, how to test them and how to interpret a gene variant, we set out to systematically review the yield of monogenic diabetes using different criteria to select people with diabetes for genetic testing and the technologies used. In addition, we sought to develop current guidelines for genetic testing for monogenic diabetes using a systematic review and grading of the studies available. The aim for this review was to fill the knowledge gaps indicated to improve diagnostics of monogenic diabetes and hence enhance the opportunity to identify those who would benefit from precision diagnostics. The evidence underpinning the link between the genetic test result and clinical management and prognostics are covered as separate systematic reviews in this series, by other members of the Precision Medicine in Diabetes

Initiative (PMDI) addressing precision treatment and prognostics for monogenic diabetes.

We provide a series of recommendations for the field informed by our systematic review, synthesizing evidence, and expert opinion. Finally, we identify major challenges for the field and highlight areas for future research and investment to support wider implementation of precision diagnostics for monogenic diabetes.

## Methods

**Registration**. We have registered a PROSPERO (International Prospective register of Systematic Reviews) protocol (ID:CRD42021243448) at link https://www.crd.york.ac.uk/prospero/. We followed the preferred reporting items for systematic reviews and meta-analysis guidelines[31].

**Search strategy**. We focused on seven questions for our review. For the questions of whom to test for monogenic diabetes, and which technologies should be used to test them, we searched PubMed (National Library of Medicine) and Embase.com using relevant keywords and thesaurus terms for relevant monogenic diabetes categories such as MODY, neonatal diabetes, lipodystrophy, mitochondrial, combined with key genes of interest (Supplementary Table 1). Publication date limitation was set to 1990-2022, human studies only and English as a language limitation. A first search was performed in October 2021 with an update in June 2022. For the remaining questions our search strategies were adapted to recognize guidelines already in place for these areas. Details of our PICOTS framework are provided in Supplementary Table 2.

**Screening**. For all questions except those relating to current guidelines, we carried out screening of papers using COVIDENCE (www.covidence.org). At least two reviewers independently screened titles and abstracts of all publications identified in the searches, blinded to each other's decisions. Conflicts were resolved by two further reviewers. All remaining articles were retrieved and screened by at least two reviewers for eligibility, recording any reasons for exclusion. Disagreements were resolved by a third reviewer.

**Inclusion/exclusion criteria**. For the question of whom to test for monogenic diabetes we included original research of any study design (cohort, case-control) but not case reports, in any human population with diabetes or mild hyperglycemia in whom the yield of monogenic diabetes was provided. A minimum of 100 unrelated probands with genetic testing results using sequencing of at least one or more genes implicated in monogenic diabetes had to be provided. Studies that only tested selected variant(s) within a gene or provided an association of common variants in monogenic diabetes genes with type 2 diabetes risk were excluded. Reviews, commentaries, editorials, and conference abstracts were excluded. Other reasons for exclusion were if studies only involved animal models or in vitro data. Studies which did not provide any diabetes screening measurements or those where the outcome was not a subtype of monogenic diabetes or those focusing on treatment response or prognosis were excluded.

For the question of which technologies should be used to test for monogenic diabetes we included original research of any study design where a genetic testing methodology was employed to diagnose monogenic diabetes in any human population with diabetes, where an evaluation of a genetic testing method had been undertaken. This included mitochondrial diabetes due to the m.3243A>G variant since this has recently been shown to be a common cause of diabetes in patients referred for genetic testing for monogenic diabetes[32]. We excluded studies using outdated or obsolete methods very rarely used by diagnostic laboratories such as single-strand conformation polymorphism analysis. Functional

studies on variants, studies detecting risk variants for polygenic forms of diabetes and linkage studies to identify candidate diabetes genes were excluded. The study had to provide a clear description of the methodology used, and studies were excluded where insufficient detail was provided.

**Data extraction**. From each included publication, we extracted data on the first author, publication year, and the following data: type of study, country, number of individuals genetically tested. For the question of who to test we also recorded reported race, ethnicity, ancestry or country of the study, proportion female to male, BMI, other characteristics of those who were tested such as age of diabetes diagnosis, or other clinical or biomarker criteria. Where available, the extracted data also included measures of diagnostic test accuracy including sensitivity, specificity, receiver operating characteristic curve, and the area under the curve for discriminating between those with monogenic diabetes and those with other etiologies of diabetes. For genetic testing methodology the number of genes tested and gene variant curation method. For all studies we recorded the number of individuals diagnosed with different monogenic diabetes subtypes, yield by different selection approaches or genetic testing technologies if applicable.

**Data synthesis**. For the question of whom to test for monogenic diabetes, we summarized the total number of monogenic diabetes studies concerning neonatal diabetes, gestational diabetes, and other atypical presentations of diabetes. For each of these presentations of diabetes we grouped them according to whether they were tested for a single gene, small (2–5 genes) or a large gene panel ≤ 6 genes. We also summarized the studies where possible by whether they included international cohorts or those that includes individuals of predominantly European or non-European descent. If self-reported race, ethnicity or ancestry information of the population was not provided, those studies conducted in countries with predominantly non-European populations were allocated to the latter group.

**Critical appraisal and grading the certainty of evidence**. A ten-item checklist for diagnostic test accuracy studies[33] was used to assess the methodological quality of each study by two critical appraisers, and any conflicts were resolved by a third reviewer for Questions 1 and 2. This tool is designed to evaluate the risk of bias relating to diagnostic accuracy studies using three items regarding patient selection and seven items regarding the index test. Patient selection items included whether there was a consecutive or random sample of patients enrolled (Item 1). This was interpreted as yes if the cohort described consecutive enrollment or a random sample from any given collection of individuals. For items 4-8, the index test was defined as the clinical features or biomarkers used to select individuals for genetic testing. The genetic test was considered the reference test, of which the current reference standard was decided to be at least a six-gene panel, including the genes most associated with the phenotype category. This for the neonatal diabetes phenotype category was considered to include *ABCC8, KCNJ11, INS, GCK, EIF2AK3, PTF1A*, and for non-neonatal beta-cell monogenic diabetes category was considered to include *GCK, HNF1A, HNF4A, HNF1B, ABCC8, KCNJ11, INS* and m.3243A>G. The reference standard genetic test for diabetes associated with a lipodystrophy phenotype category was considered to include at least *PPARG* and *LMNA*. Item eight, regarding an appropriate interval between the index test and the reference test to ensure that the status of the individual could not have meaningfully changed, was deemed not applicable to monogenic diabetes as the genetic test result remains stable throughout the person's lifetime, hence a total of 9 items of this checklist were scored for each paper. We then synthesized the

data from tabulated summaries and assessed the certainty of evidence by using the GRADE approach[34].

The GRADE approach for diagnostic tests and test strategies was applied to answer the clinical question of who with diabetes should be offered the reference genetic test if we could not afford to provide this to everyone. The aim of the test (i.e., the clinical features and/or biomarkers) was to perform a triage function for selecting those with diabetes who had a greater likelihood of having a monogenic diabetes etiology, which when correctly diagnosed would enhance their clinical management. In assigning levels of evidence to the included studies considering various triage tests, 5 criteria were used as per the Canadian guidelines for grading evidence for diabetes studies[35]. Firstly, independent interpretation of the triage test results, without knowledge of the diagnostic standard (reference genetic test result) which was item 4 of the risk of bias tool. This was considered to always be the case, given that clinical features and laboratory biomarkers (triage tests) were assessed independently of the genetic testing and variant curation (reference test). Secondly, item 7 of the risk of bias tool, independent interpretation of the diagnostic standard (the reference genetic test result) without knowledge of the triage test result, was also considered to always be the case because those instances in which interpretation of the genetic test was done with more detailed knowledge of the clinical features, could not be gleaned from the papers. Whilst gene variant curation often relies on knowledge of the clinical features and laboratory biomarkers, this criterion was not deemed sufficiently informative for decisions about grading the evidence for the question of whom to offer genetic testing for monogenic diabetes. Thirdly, the selection of people suspected (but not known) to have the disorder was considered for the summary of the evidence and related to item two of the bias tool of avoiding a case-control design. Fourthly, a reproducible description of the test and diagnostic standard was considered. Finally, at least 50 patients with and 50 patients without monogenic diabetes was a key criterion that was considered. This criterion was incorporated into the inclusion criteria for studies considered relevant for the question of whom to test, by having a minimum of 100 unrelated probands with genetic testing results. However, depending on the selection criteria used, the genetic etiologies tested for and the size of the study, the number of patients who were confirmed as having monogenic diabetes did not always exceed 50 patients. To derive the overall level of evidence for the published studies for any group of triage tests, all five criteria had to be present for level 1, four criteria for level 2, three criteria for level 3 and one or two criteria for level 4 evidence. We developed guideline recommendations for whom to test for monogenic diabetes by assigning grade A for those criteria that were supported by best evidence at level 1, grade B for those that were supported by best evidence at level 2, grade C for those that were supported by best evidence at level 3 and Grade D for those that were supported by level 4 or consensus. Details of our pipeline for assessing the level of evidence and grade are outlined in Supplementary Figure 2.

Answering Questions 3 (*On what basis is a gene considered a cause of monogenic diabetes*), 4 (*On what basis is a variant considered a cause of monogenic diabetes*), and 5 (*How should a gene variant causing monogenic diabetes be reported*) are central to putting knowledge about monogenic diabetes etiology into practice. Currently, individual laboratories select the genes to include on NGS panels, interpret variants according to internal guidelines, and create reports based on internal procedures. Recognizing the need for clarity and consistency in these areas, several national and international guidelines have been developed and refined. It was recognized that several general resources exist for assessing whether a gene is implicated in a disease, including the crowd-sourced UKPanelApp[36] and the ClinGen evidence-based Gene-Disease Validity framework[37]. It was also noted that the ClinGen MDEP GCEP (https://clinicalgenome.org/affiliation/40016/) has convened to apply the ClinGen evidence-based framework to monogenic diabetes. Therefore, a de novo systematic evidence review for this question was not considered necessary or useful for this document, but rather a description of these existing resources and how they can be accessed. Similar to question 3, for question 4, it was recognized that consensus guidelines for assessing the role of specific genetic variants in disease were issued jointly by the American College of Medical Genetics and Genomics (ACMG) and the Association for Molecular Pathology (AMP) in 2015[38] and the Association for Clinical Genomic Science in 2020. The ACMG/AMP guidelines have been expanded and refined by ClinGen[38–42], and the ClinGen MDEP VCEP (https://clinicalgenome.org/affiliation/50016/) has convened to develop gene-specific rules for applying the guidelines to monogenic diabetes. For reporting genetic testing results (Question 5), there are some general published consensus guidelines[38,43,44], and a limited emerging literature reporting studies evaluating report utility[45] that was deemed not sufficient for a systematic evidence review. In this document, these are summarized and recommendations specific to monogenic diabetes are proposed based on existing practice.

For our evaluation of the next steps after a diagnosis of monogenic diabetes (Question 6), we excluded articles that either did not answer the question or only included a cursory general mention of the value of genetic testing for management. We reviewed the remaining 36 publications, consisting of specific case studies, cohorts, and review articles. Twelve papers discussed monogenic diabetes testing and/or treatment in adults and children. Seven articles described strategies for testing and/or management of monogenic diabetes during pregnancy. Three articles focused on maternally inherited diabetes and deafness (MIDD), five centered on neonatal diabetes, and nine covered syndromic forms of monogenic diabetes, including Wolcott-Rallison, Alström, and Wolfram syndromes. We then reviewed the literature for additional published studies relating to the steps after monogenic diabetes diagnosis. Information from publications was combined with expert advice from genetic counselors and physicians who specialize in monogenic diabetes clinical care. This section includes recommendations for results disclosure, cascade testing and addressing non-medical issues that may arise. We direct the reader to other systematic reviews in this series for prognostics and treatment recommendations. To evaluate the challenges for diagnostic testing for monogenic diabetes (Question 7) we screened 455 abstracts for challenges for the field of monogenic diabetes diagnosis of which 41 were screened as full text articles and 14 taken forward for full text extraction.

**Reporting summary**. Further information on research design is available in the Nature Portfolio Reporting Summary linked to this article.

## Results

**Question 1—Who to test for monogenic diabetes?** For the question of who to test for monogenic diabetes, a total of 12,896 records were retrieved. In Covidence, 2,430 duplicates were identified. We included 100 publications from 10,469 publications screened (Supplementary Fig. 1A). The key data from each of the 100 studies were included in Supplementary Data 1 and the 10-item checklist assessments for these papers were summarized in Fig. 1A. The summary of evidence from the included studies is detailed in Tables 1–2. We also provide a list of recommendations based on this evidence in Table 3.

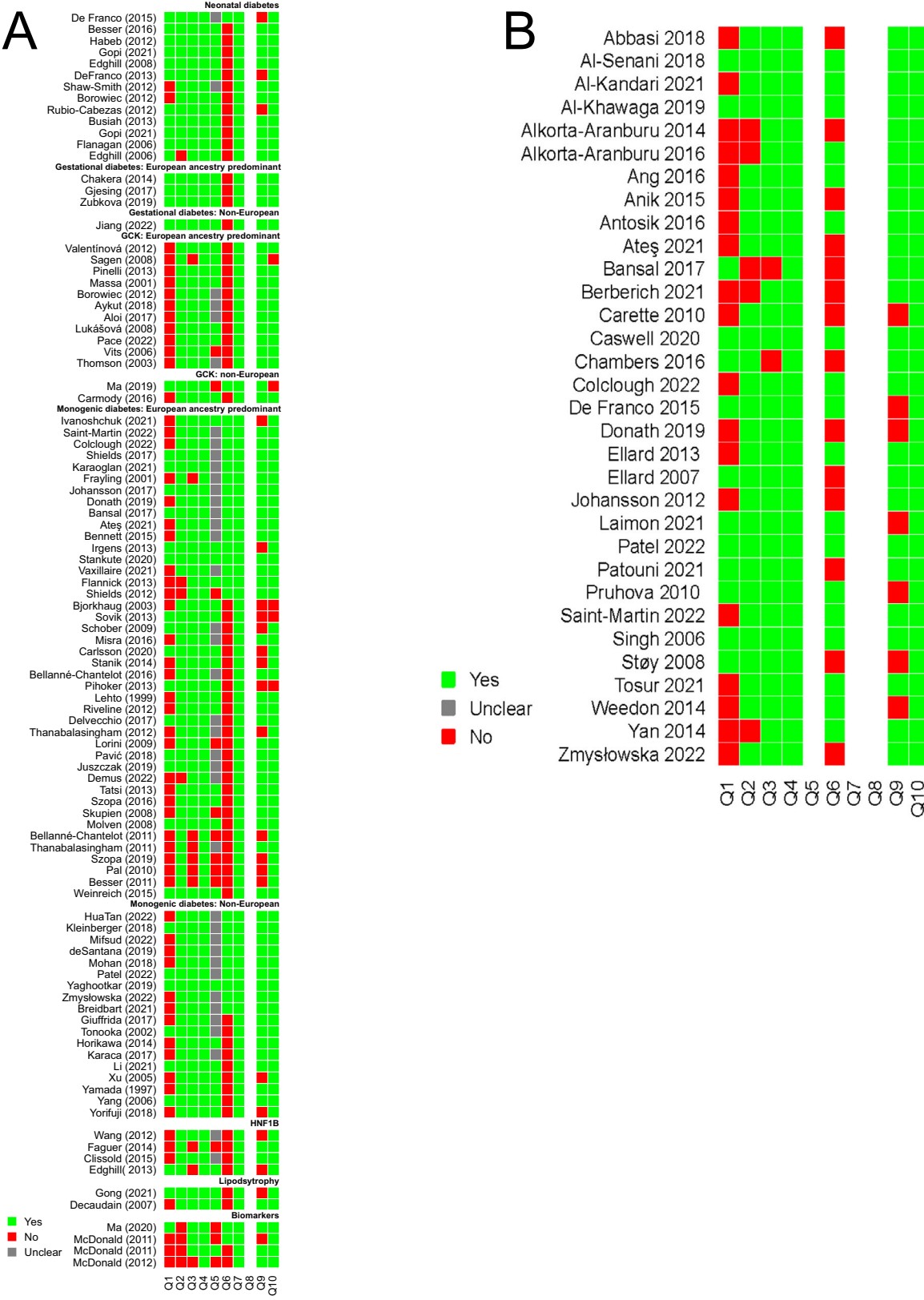

**Fig. 1 Critical appraisal of evidence using the Joanna Briggs Institute (JBI) Critical Appraisal tool for Systematic Reviews. A** Results for papers from question 1. **B** Results for papers from question 2. The horizontal axis for the heatmap refer to questions 1–10 of the 10-item JBI checklist. Green is Yes, Red is No, Gray is unclear. Non-applicable answers were left blank. A completes list of papers in (**A**, **B**) can be found in Supplementary Data 1 and 2.

**Table 1 Important summary of findings from sequencing studies for monogenic diabetes.**

| Diabetes population studied (country/ancestry of population) | Genetic testing methodology | Number of studies (Range of sample size tested) | Yield by key characteristics of diabetes population tested | Level of evidence |
|---|---|---|---|---|
| Neonatal diabetes diagnosed <6 months (International) | >5 genes (23 gene panel) | 1 study (n = 1020) | **Neonatal diabetes diagnosed < 6 months tested using large-gene panel has high yield for monogenic etiology.** Yield in a large, unselected other than by age at diagnosis, international study: 840/1020 (82%) | 1 |
| Neonatal diabetes diagnosed <6 m (UK, Saudi Arabia, India) | ≤5 genes including *KCNJ11*, *ABCC8* | 3 studies (n = 165–750) | **Neonatal diabetes diagnosed < 6 months tested with small gene panel including KCNJ11/ABCC8 has high yield for monogenic etiology.** UK:598/750 (78%), Saudi/UK: 56/88 (64%) Saudi, 32/77 (42%) British India: 39/181 (22%) | 1 |
| Neonatal diabetes diagnosed <6 m (International cohorts) | Single genes: *INS*, or *SLC19A2* or after excluding *INS*, ABCC8, *KCNJ11* then *PDX1* or *GCK* | 4 studies (n = 103–212) | **Neonatal diabetes diagnosed < 6 months tested for single genes with or without excluding more common gene etiologies have lower yields.** INS: 33/141 (23%) SLC19A2: 3/212 (1%) PDX1: 3/103 (3%) GCK homozygous: 1/17 (6%) | 1 |
| Neonatal diabetes diagnosed <12 m (Spain, France, India) | ≤5 genes Including *INS* or *KCNJ11* and *ABCC8* | 3 studies (n = 189–405) | **Neonatal diabetes diagnosed <12 months tested for common genes has a lower yield than for those diagnosed < 6 months and there are less cases.** Spain: 263/405 (65%) diagnosed <6 m, and 9/145 (6%) diagnosed 6–12 m France: 64/155 (41%), diagnosed <6 m and 5/18 diagnosed 6–12 m (28%) India: INS only in PNDM, Ab-ve, CP+ve, diagnosed<9 m: 8/189 (4%) | 1 |
| Neonatal diabetes diagnosed <24 months (UK, International samples) | *KCNJ11* only or *INS* after *KCNJ11* negative | 3 studies (n = 58–70) | **There were no cases of monogenic diabetes found in children diagnosed age 12-24 months although only limited genes were tested.** 0/70 KCNJ11 0/63 KCNJ11 0/58 INS | 2 |
| Gestational Diabetes (GDM) European cases | GCK only | 3 Studies (n = 188–400) | **In European women with GDM, yield for GCK-hyperglycemia was 1–6% when otherwise unselected, rising to 22% when only women without obesity were selected for testing.** -UK and Ireland: overall 4/356 (1%) -Diet-treated Danish GDM: 21/354 (6%) mean BMI 28 -Non-obese Russian GDM: 41/188 (22%) **Selecting by fasting glucose of above 5.5 mmol/L increased yield:** ~ GCK detected in 0/109 women with GDM and fasting gludose <5.1, 1/129 women with GDM and fasting glucose 5.1-5.5, and 3/118 women with GDM and FG > = 5.5 **Combination of fasting glucose and BMI improved yield:** -BMI = < 21 and FG > = 5.5 sensitivity 20%, specificity 100% -BMI < 25 and FG > = 5.5, sensitivity 68%, specificity 99% -BMI < 30 and FG > = 5.5, sensitivity 82%, specificity 96% | 2 |
| Gestational Diabetes (GDM) China | GCK only | 1 Study (n = 411) | **There is a lack of studies in non-European women to define the best testing criteria for GCK-hyperglycemia in pregnancy.** In Chinese women with GDM the yield for GCK was 4% (15/411) when otherwise unselected | 3 |
| Children and adults with diabetes, not GDM. | GCK only | 12 Studies (n = 100–722) | In non-GDM European studies, the yield for GCK-hyperglycemia in those with persistent, stable, mild hyperglycemia or fasting hyperglycemia is high: 30- | 1 |

**Table 1 (continued)**

| Diabetes population studied (country/ ancestry of population) | Genetic testing methodology | Number of studies (Range of sample size tested) | Yield by key characteristics of diabetes population tested | Level of evidence |
|---|---|---|---|---|
| Predominantly European ancestry | | | 74%.<br>-criteria used for mild fasting hyperglycemia FG 5.5-7.8 mmol/L<br>-criteria used for mild hyperglycemia HbA1c 38-62 mmol/mol<br>Adding other criteria such as absence of obesity or family history of diabetes does not consistently increase the yield (27-88%). | |
| Children and adults with diabetes, not GDM. Predominantly non-European ancestry | GCK only | 3 studies (n = 24 and 679) | In a non-GDM Chinese study, those with mild fasting hyperglycemia and low triglycerides, had low yield for GCK (compared to European studies):<br>-Discovery group 11/545 (2%) and replication groups 1/207 (0.5%)<br>In a US study, those with persistent, mild fasting hyperglycemia plus FH or BMI < 30 or diagnosis <30 years showed high yield 64/117 (55%), with similar results in Latin and Asian subgroups. | 2 |
| Children and adults with diabetes, not GDM. Predominantly European ancestry | Large monogenic diabetes panels (5–28 genes) | 16 studies (n-178-6888) | **Yield in Europeans using large MD gene panel varies by age of diabetes diagnosis and other selection criteria from 0.7% to 34%, and low yield in unselected adults with older age of diabetes diagnosis.**<br>▪ In suspected MODY cohorts yield was 16%-23% (4 studies)<br>▪ In the Norwegian Childhood Registry, yield in children diagnosed **<15 years** was 19/462(4%, including 1 who was antibody positive)<br>▪ In Turkish children diagnosed **<18 years** atypical for T1D or T2D, yield was 24/330 (10%, including 4 who were antibody positive)<br>▪ In a UK study, those Ab-ve diagnosed **<30 years** and either C-peptide positive or non-insulin treated, yield was 4% (51/1407), and higher at 21% (38/178) in additionally non-obese French individuals diagnosed **<35 years**<br>▪ In a French study the comparative yield in those with (a) diabetes diagnosed 15-40 years, BMI < 30 and FH was 198/999 (20%) vs (b) any two of these criteria was 254/1564 (16%) vs (c) Caucasian plus (a) was 54/504 (11%) vs (d) diabetes diagnosis **<40 years**, BMI < 25 was 101/301 (34%).<br>▪ In a German study of people with Ab-ve and diabetes diagnosed>40 y was 0.7% (18/2670) vs diabetes diagnosed **<40 years** was 2% (29/1346) | 1 |
| Children and adults with diabetes, not GDM. Predominantly Non-European ancestry | Large MD gene panel (>5 genes) or sequential targeted exome/ whole exome sequencing | 6 studies n = 184-488 | **Yield in mixed ethnicity cohorts using large MD gene panel was similar to that seen in the European cohorts at 13%-26% using various selection criteria, but with higher autosomal recessive etiologies in populations with higher consanguinity:**<br>▪ Turkish children diagnosed **5-12years**, either low T1D GRS or mod T1D GRS with Ab-ve had yield 34/236 (14%), 14 of the 34 were autosomal recessive diseases (~ 20% consanguinity)<br>▪ USA/India cohort diagnosed **<30 years**, Ab-ve, FH: 39/152 (26%)<br>▪ France cohort Ab-ve and 2 or more of diagnosed<40 y, BMI < 30 at diagnosis, FH: 315/1975 (16%)<br>▪ Suspected MODY cohorts 13% and 15% in Brazil and Singapore 2 | 1 |

**Table 1 (continued)**

| Diabetes population studied (country/ ancestry of population) | Genetic testing methodology | Number of studies (Range of sample size tested) | Yield by key characteristics of diabetes population tested | Level of evidence |
|---|---|---|---|---|
| Small MD panels or 3–5 individual genes | 7 studies $n = 100$–4010 | **Yield from testing 3–5 common MD genes varies widely by age and selection criteria (from 8%–97%):** ∎ Suspected MODY (4 studies) ranged from 13% in South Asians, 29% in other UK cohort, 40% in French, 57% in Slovakia/Czech, to 97% in German study. ∎ In those diagnosed 1–18 years, 4 comparative selection criteria yields ranged from 0% (0/182) in Ab+ve, vs 15% (46/303) in Ab−ve, vs 30% (29/96) with additional FH vs 34% (44/131) with additional FH or HbA1c < 7.5% at diagnosis ∎ The USA SEARCH cohort with high prevalence of obesity and non-European ancestry, diagnosed <20 years, Ab−ve, C-pep +ve, had yield 8% | | |
| Children and adults with insulin resistance and lipodystrophy | Individual genes | 2 studies $n = 277$–1002 | **Yield from testing varied between 0.6% for PPARG and 9.7% for LMNA** ∎ Han Chinese population 1002 patients recruited between 2014–2018 with T2D diagnosed 18–40 years, autoantibody negative, C-peptide positive sequenced for PPARG. 6/1006 (0.6%) ∎ French cohort of 277 probands recruited between 2002–2006 with suspected severe insulin resistance, lipdystrophy and/or android body habitus, insulin restance or altered glucose tolerance. Sequenced for LMNA mutations 27/277 (9.7%) | 3 |

**Table 2 Diagnostic screening strategies for monogenic diabetes using clinical risk score, decision trees and novel biomarkers.**

| Type of study | Study population | Summary | Performance |
|---|---|---|---|
| **HNF1B risk score** | Faguer (2014)[115]<br>433 French individuals who had a variety of structural developmental anomalies (renal, pancreatic, liver and GU) including 56 with known HNF1B<br>Clissold (2015)[62]<br>686 individuals with congenital kidney anomalies referred to the UK monogenic diabetes testing service for HNF1B genetic testing, of whom 177 (26%) had an HNF1B mutation or deletion | A weighted 17-point HNF1B score using mainly renal and urinary tract structural features, was derived from literature searches, and evaluated in a discovery cohort by Faguer et al, then tested by Clissold in an independent cohort of cases referred for HNF1B genetic testing.[62,115] | A score of 1-4 points was possible for each feature. Median score for the discovery group was 8 (0-24), with a higher median score of 12 (6-22) in HNF1B.<br>● The ROC curve C-statistic was 0.78 for a score of <8 to predict negative testing for HNF1B, with sensitivity 98.2%, specificity 41%, PPV 20%, NPV > 99%<br>In the independent test cohort, using a threshold score of 8<br>● sensitivity of 80%, specificity of 38%, PPV of 31% and NPV of 85%<br>The performance showed reduced clinical utility in the UK cohort, which would lead to missed cases, probably reflecting underlying differences in the patient group referred. |
| **"MODY" risk calculator** | Shields (2012)[63]<br>1191 individuals including with known monogenic diabetes (296 HNF1A, 243 GCK, 55 HNF4A), 278 T1D and 319 T2D diagnosed up to age 35 years only. | A web based "MODY" probability model using 8 simple clinical features, derived by logistic regression comparing known HNF1A, GCK, HNF4A and gold standard T1D and T2D groups. | Any MODY vs T1D: ROC C-statistic 0.95, 87% sensitivity and 88 specificities for a probability of 40%.<br>Any MODY vs T2D: ROC C-statistic 0.98, sensitivity 92%, specificity 95% for a probability of 60% |
| **Monogenic diabetes screening strategies using clinical features and laboratory biomarkers** | Shields et al (2017)[116]<br>1418 individuals diagnosed with diabetes up to age 30 years from 2 UK regions. 34 known and another 17 new cases of monogenic diabetes were identified. | Participants were initially screened to find antibody negative and urinary C-peptide positive cases (n = 216) who went on to have genetic testing to estimate the prevalence of monogenic diabetes in each region. 8 common etiologies identified using Sanger sequencing and 9 rare causes identified using next generation sequencing. | The diagnostic yield was 51/216 (24%) comprising 17 new cases.<br>The positive predictive value of the pathway was 20%, negative predictive value 99.9% |
| **HNF1A diabetes screening strategies using clinical features and laboratory biomarkers** | Ma (2020)[117]<br>Chinese individuals with young-onset diabetes diagnosed <age 40 years: The yield of HNF1A-diabetes was 9/410 (2.2%). | A clinical screening strategy to identify those most likely to have HNF1A-diabetes was devised using 2 discovery cohorts and 4 criteria were selected: BMI <28Kg/m$^2$, HDL-chol >1.12 mmol/L, Fasting insulin <102pmol/L and hsCRP <0.75 mg/L<br>These criteria were assessed in a test cohort where T1D had been excluded. | Two clinical screening strategies were tested:<br>CSS1 tested individuals with any 3 out of 4 of the criteria (n = 131):<br>● sensitivity 100%, specificity 69.7%,<br>● positive and negative predictive values of 6.9% and 100%.<br>CSS2 tested those with all 4 criteria (n = 51)<br>● sensitivity 88.9%, specificity 89.6%,<br>● positive and negative predictive values of 15.7% and 99.7%.<br>Addition of the biomarkers reduced the number of individuals who needed genetic testing by 68-88%, without significant reduction in cases identified. |
| | Bellanne-Chantelot (2016)[118]<br>668 individuals, 143 with known HNF1A-diabetes, 301 Young T2D with suspected HNF1A-diabetes and 215 with T2D | Models using clinical parameters (gender, ethnicity, family history, diagnosis age, BMI and HbA1c) and the addition of hsCRP were compared to predict HNF1A-diabetes | The ROC C-statistic to distinguish HNF1A- from YT2D was 0.82 for clinical features and 0.87 when including hsCRP. However use of hsCRP did not have a significant diagnostic added value in a "gray zone" analysis, where 65-68% of cases were in a zone of diagnostic uncertainty using either clinical features alone or adding hsCRP |
| | Juszczak (2019)[64]<br>989 European individuals with antibody negative, c-peptide positive diabetes diagnosed before age 45 years. All patients had HNF1A sequenced and 16/989 (1.6%) had HNF1A-diabetes based on strict pathogenicity criteria. | The utility of the HNF1A biomarkers hsCRP and fucosylated glycan (GP30) were assessed. | Glycan GP30 had a ROC C-statistic = 0.90 (88% sensitivity, 80% specificity, cutoff 0.70), and hs-CRP had a ROC C-statistic = 0.83 (88% sensitivity, 69% specificity, cutoff 0.81 mg/L) and reduced the number of cases needed to sequence by 70-75%. |

In neonatal diabetes there were a total of 13 studies, of which three included those diagnosed with diabetes within 24 months of age, three within 12 months of age and the rest within six months of age (Table 1). There was only one study which used the reference standard large gene panel for neonatal monogenic diabetes diagnosis, while the rest did not. The highest yield of 82% was obtained in a single international cohort study of 1,020 patients diagnosed with diabetes within six months of age using a large 23-gene panel[46]. Of these, 46% had *KCNJ11* or *ABCC8* followed by *INS* as the next common etiology.

For neonatal diabetes diagnosed between 6–12 months the yield was 0-28% derived from six studies containing sample sizes of 18 to 145 individuals tested using only *KCNJ11, ABCC8, INS genes*. No cases of monogenic diabetes were found in the small subpopulations tested with diabetes diagnosed 12-24 months in three studies sequencing *KCNJ11 and INS* genes only (n = 58–70). The risk of bias criterion for patient selection was high for one study in which a random sample of patients had not been enrolled[47]. Two studies were deemed to be at risk of bias due to not all receiving the reference test[48,49] (Fig. 1A).

**Table 3 Recommendations based on the synthesis of evidence for who to test for monogenic diabetes.**

| Recommendation | Who to test for monogenic diabetes | Grade |
|---|---|---|
| 1 | All patients diagnosed with diabetes before the age of 6 months should be tested for monogenic forms of neonatal diabetes using the large-gene panel. | A |
|  | All patients diagnosed between 6 and 12 months should be tested for monogenic forms of neonatal diabetes using the large-gene panel. No demonstrable yield of monogenic etiology to support reflexive genetic testing patients diagnosed with diabetes between 12-24 months. | B |
| 2 | Women with gestational diabetes and fasting glucose above 5.5 mmol/L without obesity* should be tested for GCK etiology. | B |
| 3 | Those with persisting, mild hyperglycemia (HbA1c 38–62 mmol/mol, or fasting glucose 5.5-7.8 mmol/L) at any age, in the absence of obesity* should be tested for GCK etiology. | A |
| 4 | People without obesity under the age of 30 years who are either autoantibody negative and/or have retained C-peptide levels should be tested for monogenic diabetes using a large-gene panel | A |

*by selecting those who are of normal weight or underweight (rather than those who are non-obese) to offer genetic testing to, increases the specificity but reduces the sensitivity for detecting GCK and may be considered a less costly approach.

The selection by age of diabetes diagnosis between six and 12 months for neonatal diabetes genetic testing was supported by level 1 evidence from 1 large study and thereby supports this being a Grade A recommendation (Table 3). Selection by age of diabetes diagnosis between 6 and 12 months was supported by a yield of 6% to 28% by level 2 evidence from six studies, although these were limited by only testing for *INS* or *KCNJ11* and *ABCC8*. Selection by age of diabetes diagnosis beyond 12 months for monogenic diabetes testing was not supported by three studies examining those diagnosed with diabetes up to 24 months. These failed to find any cases of monogenic diabetes although these were limited by only testing for *KCNJ11* or *INS* etiologies in small cohorts with diabetes diagnosed between 12-24 months (level 2 evidence).

In gestational diabetes mellitus (GDM), the recommendation that all women without obesity should be tested for GCK etiology (Table 3) were derived from a total of four studies which examined *GCK* diagnosis only, of which three were in predominantly European women[50–52], and one study was in Chinese women[53] (Table 1). The yield for *GCK* etiology ranged from 1–6% in otherwise unselected women with GDM, however, increased to 22% when only non-obese women were selected for *GCK* testing[52]. The use of fasting glucose of 5.5 mmol/L or higher was demonstrated to have the highest yield (3/118) for GCK, with only 1/129 women with fasting glucose below 5.5 mmol/L found to have GCK etiology and 0/109 women with fasting glucose below 5.1 mmol/L[50]. In this 2014 study the trade off between specificity and sensitivity of detection for GCK among the women with GDM (and fasting glucose above 5.5 mmol/L), by various BMI thresholds is shown (Table 1). Given none of these studies included over 50 women with *GCK* who had been suspected but not known to have the disorder in the study, the highest level of evidence was graded at level 2. Other than only testing for the single gene *GCK*, there were no other concerns about bias in these studies.

For *GCK* testing in those without GDM, the recommendation to provide this for those with persisting, mild hyperglycemia at any age, in the absence of obesity (Table 3) is based on a total of 13 studies of which 11 were in predominantly European populations (Table 1). Overall, there was frequent assessment of bias in patient selection criteria used in all but one study. There were 5 studies with either unclear or no defined thresholds provided for persistency or stability of mild hyperglycemia using various fasting glucose and or HbA1C thresholds. The yield for *GCK* etiology ranged from 0% in unselected cases of hyperglycemia and increased to 30-74% in those with persistent, stable, mild hyperglycemia (Table 1). There was only one Italian study of 100 individuals that compared two testing strategies[54]. This study demonstrated that the yield for *GCK* increased in those with impaired fasting glucose and without diabetes autoantibodies from 32% when one MODY criteria was added compared to 88% when non-obese and lack of diabetes medications were added. However, this study characteristics provided level 3 evidence. The yield for GCK-etiology in a Chinese study which used mild, fasting hyperglycemia and low triglycerides was relatively low (2% vs 0.5% in discovery and replication datasets of $n = 545$ and $n = 207$ respectively)[55]. However, in a mixed ethnicity population in the USA, selection of those with persistent, mild fasting hyperglycemia plus either family history or BMI below 30 kg/m$^2$ or diabetes diagnosis age below 30 years produced a yield of 55%[56]. Overall, four studies supported level 1 evidence for selecting those with persisting, mild, fasting hyperglycemia for *GCK* testing.

The recommendation to provide monogenic diabetes testing to people without obesity under the age of 30 years who are either autoantibody negative and/or have retained C-peptide levels was derived from a total of 60 studies. These examined the yield of monogenic diabetes beyond the neonatal period, of which 43 were in predominantly European populations. Of these, 25 studies utilized the reference standard of the large-gene panel (8 in non-European populations). The yield varied by the triage test strategy utilized to select individuals for genetic testing and those receiving the large-gene panel had a greater yield than smaller or single-gene approaches. Younger age of diagnosis of diabetes (thresholds included below 15, below 18, below 25, below 35 and below 40 years) and negative diabetes autoantibodies were the most common triage test strategy. Excluding those with type 1 diabetes using either negative diabetes autoantibodies or presence of C-peptide or both was frequently employed. With the large-gene panel approach, the yield for a monogenic etiology ranged from 0.7% to 34%. There was low yield of 18/2670 (0.7%) in those with negative antibodies who had diabetes diagnosed above the age of 40 years[57]. In suspected MODY cohorts, the yield was 16% to 23% (Table 1). Most of such studies were assessed as having bias in patient selection and many did not have a clear description of "suspected MODY" (Fig. 1B). One French study of 1564 individuals provided the comparative yield for (a) 3 clinical criteria of diabetes diagnosis age of 15–40 years, BMI below 30 kg/m$^2$, and family history of diabetes which was 20% vs (b) for any 2 of these clinical criteria the yield was 16% vs (c) diabetes diagnosis of 15–40 years and BMI below 25 kg/m$^2$ the yield was 34%[58]. In a Turkish cohort of children with diabetes (diabetes diagnosis age IQR 5–12 years), with either low Type 1 diabetes genetic risk score (T1GRS) or moderate T1GRS and negative diabetes autoantibodies the yield was yield of 34/236 (14%). This included 14/34 autosomal recessive cases, with approximately 20% prevalence of

consanguinity in the tested population[59]. While there was considerable heterogeneity in selection criteria used, the best evidence was at level 1 for selecting those diagnosed with diabetes below the age of 30 years who are either autoantibody negative/ and or have retained C-peptide (for lowering probability of type 1 diabetes) and those without obesity (for lowering probability of type 2 diabetes), for testing for monogenic diabetes using the reference large gene panel.

In people with diabetes, there were 2 studies which evaluated the yield for PPARG or LMNA etiology. There was low yield of 0.6% for PPARG etiology in a Chinese study of people diagnosed with diabetes between 18–40 years who were antibody negative and C-peptide positive[60]. However, in those selected on the basis of lipodystrophy and/or severe insulin resistance, the yield was 9.7% for LMNA etiology in a French study[61].

For HNF1B yield in populations selected for diabetes, there were 4 studies (Supplementary Data 2, Fig. 1B). A yield of 2.4% and 2.9% for HNF1B etiology for diabetes was found in a UK and Chinese study respectively. In the UK study, those referred for monogenic diabetes who had no other common etiology detected received HNF1B sequencing and doseage analysis. In the Chinese study, older adults with diabetes who were antibody negative and had either renal structural abnormalities or impaired renal function, were tested.

Besides the yield of monogenic diabetes using simple clinical criteria, the sensitivity and specificity of various clinical risk scores and biomarkers are summarized in Table 2. The clinical risk scores have been derived for either a single genetic etiology (eg: HNF1B risk score)[62,63] or a group of genetic etiologies collectively (eg: "MODY risk calculator" for subtypes GCK, HNF1A, HNF4A)[63]. The utility of routine biochemical biomarkers such as hsCRP or antibodies have been investigated for distinguishing several genetic etiologies (HNF1A, HNF4A, HNF1B) from Type 1 or Type 2 diabetes, or for distinguishing HNF1A specifically from Type 2 diabetes using HDL-cholesterol or hsCRP[64–69]. Low T1GRS has been used as a triage test in addition to negative diabetes antibodies for selecting individuals for broad monogenic diabetes panel testing[59,70]. Other non-routine biomarkers such as lipid fractions and glycan moieties regulated by HNF1A have been explored for distinguishing HNF1A from other diabetes subtypes, but these have added complexity and cost above clinical features[71,72], without better informing whom to test for the greater yield of monogenic diabetes provided from large-gene panel testing.

The use of 8 simple clinical features in the logistic regression model based MODY risk calculator distinguished the 3 common monogenic diabetes subtypes GCK, HNF1A, HNF4A collectively from Type 1 diabetes or Type 2 diabetes with a c-statistic of 0.98 and 0.95 respectively. This indicates excellent discrimination between Type 1 diabetes or Type 2 diabetes and the 3 common monogenic diabetes subtypes GCK, HNF1A, HNF4A collectively, as well as when comparing Type 1 and Type 2 diabetes with HNF1A/4A and GCK etiologies separately. However, limitations of this calculator include lack of validation for non-European poulations, those diagnosed with diabetes above 35 years, those with other forms of monogenic diabetes, and weaker performance in insulin-treated patients where the probability of type 1 diabetes is high. The use of negative antibodies and detectable C-peptide biomarkers to exclude Type 1 diabetes and select individuals for monogenic diabetes testing had higher yield than the use of MODY probability calculator pre-test probability of >25%. The latter had a high PPV (40%), but missed more cases (55%) compared with the antibody/C-peptide biomarker pathway (PPV 20%). Traditional MODY criteria (age at diabetes diagnosis less than 25 years, non-insulin requiring and a parent affected with diabetes) had a PPV of 58%, yet missed even more cases (63%) compared with the biomarker pathway.

**Question 2—How to test for monogenic diabetes?** For the question of which technologies should be used to test for monogenic diabetes, we included 32 studies from 2,102 publications screened (Supplementary Fig. 1B). A total of 32 studies which accessed 76 different genes were analyzed (Supplementary Data 2, Table 4) and assessed for methodological quality (Fig. 1B). NGS was the most used technique, with 16/22 NGS studies using a targeted panel. Where NGS was employed, the monogenic diabetes diagnostic yield increased by around 30% compared to Sanger sequencing of GCK, HNF1A and HNF4A alone, and resulted in the (often unexpected) diagnosis of rare syndromic forms of diabetes, most commonly m.3243A>G. NGS technologies also enabled the diagnosis of multiple monogenic subtypes in the same patient, and diagnosed patients who were missed by previous Sanger sequencing due to allelic drop-out. Gene agnostic exome and genome strategies were rarely used and did not increase diagnostic yield. Copy-number variant (CNV) analysis (by Multiplex-Ligation-Dependent Probe Amplification [MLPA] or NGS) increased diagnostic yield mostly by detecting HNF1B deletions. Non-coding variants were rare but important findings and required genome sequencing or specific targeting of non-coding mutation loci. A high diagnostic yield (74%) was reported when performing Sanger sequencing of GCK in patients with a clinical suspicion of GCK due to persistent, mild, fasting hyperglycemia. Similarly, variants in KCNJ11, ABCC8 and INS accounted for 50% of neonatal diabetes mellitus (NDM) cases and were sequenced by Sanger first in some studies. 6q24 abnormalities were also a common cause of NDM and required a specific methylation-sensitive assay to detect them. Recessively inherited and syndromic forms of monogenic diabetes were predominant in countries with high rates of consanguinity. Real-time PCR and pyrosequencing were highly sensitive and specific techniques for detecting m.3243A>G and quantifying heteroplasmy, and ddPCR successfully determined all fetal genotypes in a cell-free fetal DNA prenatal testing study of 33 pregnancies. Based on our systematic review of the literature we can make several recommendations (Table 5).

**Question —What is the basis for considering a gene as a cause of monogenic diabetes?** A general evidence-based framework for evaluating gene disease validity has been developed by the ClinGen and published by an inter-institutional group of clinical and molecular genetics and genomics experts[37]. This framework involves evaluating case level, segregation, and functional data for previously reported variants and functional data for the gene itself to classify gene-disease validity relationships into Definitive, Strong, Moderate, Limited, Disputed, or Refuted categories based on a point system combined with expert consensus for the final assignment. Tools for implementing this are available at the ClinGen website. The international MDEP GCEP has convened with the goal of curating gene-disease validity for monogenic diabetes genes and has completed the common genes (https://www.clinicalgenome.org/affiliation/40016/) and is working on expanding beyond these genes. Other general repositories for gene-disease validity curation include the crowd-sourced Genomics England PanelApp[36]. For monogenic diabetes, a curated list of monogenic diabetes genes is available at the website for the University of Exeter, where most of the research and clinical monogenic diabetes testing for the UK is conducted (https://www.diabetesgenes.org/).

Over recent years the increased availability of high throughput sequencing has led to a substantial increase in the number of genes reported to cause monogenic diabetes. The evidence that supports these gene-disease relationships does, however, vary widely. Whilst there is overwhelming genetic evidence that

**Table 4 Summary of findings for testing platforms for monogenic diabetes.**

| Cohort & Method (country/ ancestry of population) | Number of studies | Range of sample size tested | Yield of Testing | Level of Evidence | Plain-Language Summary |
|---|---|---|---|---|---|
| NDM – Sanger (ABCC8, KCNJ11 & INS) (International cohorts) | 4 | 26-1020 | 49% (580/1183) | 1 | • ABCC8 and KCNJ11 are common causes of NDM and inform transfer from insulin to sulfonylurea. • Rapid diagnosis by initial Sanger is recommended, although tNGS that includes these genes as a first line test is suitable if result within 1-2 weeks. • INS may be included in first line Sanger testing given the small size of this gene. |
| NDM – Methylation status at 6q24 locus (International cohorts) | 4 | 18-1020 | 11% (125/1137) | 1 | • Abnormal methylation at the 6q24 locus causes transient NDM but detection requires a specific assay (MS-MLPA) that requires additional resources. • Testing may be offered to all newly diagnosed NDM patients after negative Sanger and/or NGS testing. • Alternatively testing may be offered only to patients with TNDM or where DM later remits to reduce cost, but this may delay time to diagnosis. |
| NDM - NGS (all known NDM genes) (International cohorts) | 8 | 7-1020 | 70% (837/1196) | 1 | • ABCC8 and KCNJ11 are the common NDM subtypes but an additional 19 different genetic subtypes were diagnosed using NGS. • NGS testing increased diagnostic yield by 30% • Rare recessive NDM syndromes were more common than ABCC8 & KCNJ11 in consanguineous populations, and NGS testing is essential; the distal enhancer of PTF1A is mutated in 3% of cases and must be specifically targeted by NGS. • NGS may detect variants missed by Sanger sequencing due to sequence variation in primer binding sites. |
| MD – Sanger (France, USA, Greece) | 3 | 84-140 | 13% (24/181) | 1 | • Sanger sequencing of the most common cause MD genes is a viable option where NGS is not available but has lower sensitivity and will miss cases. • Testing of HNF1A, HNF4A, GCK and m.3243 A > G will diagnose around 75% of genetically confirmed MD referred to a diagnostic testing laboratory. • Laboratories issuing a no diagnosis Sanger report must inform the clinician that a diagnosis of other MD subtypes has not been excluded and advise on further testing. |
| MD - CNV detection (International cohorts) | 11 | 31-1564 | 1% (63/5051) | 1 | • CNVs are a rare cause of MD. • Most common CNV is the deletion of HNF1B associated with diabetes and structural renal disease. • Use of MLPA in Sanger or tNGS negative cases gives a small increase in diagnostic yield (~1%) but may not justify the increased cost and resources. • Read depth from NGS testing can be used simply and freely to detect CNVs and is recommended. |
| MD – NGS (International cohorts) | 16 | 9-4016 | 30% (1700/5790) | 1 | • Recommended option for first line testing, especially in populations with higher levels of consanguinity. • NGS increases diagnostic yield through testing of many more genes related less common monogenic subtypes and syndromic forms. • Diagnoses MD in 20-30% of cases with a clinical suspicion. • Yield further increased when detection for CNVs and the m.3243A>G mutation is included in the NGS assay. • Targeted custom gene panels, exome or whole genome sequencing can be undertaken, although exome and genome options are costly and better suited to novel gene discovery on a research basis. |

**Table 4 (continued)**

| Cohort & Method (country/ ancestry of population) | Number of studies | Range of sample size tested | Yield of Testing | Level of Evidence | Plain-Language Summary |
|---|---|---|---|---|---|
| m.3243A>G Genotyping (UK, China) | 2 | 57–230 | 83% (47/57) | 2 | • It is possible for a patient to be diagnosed with more than one monogenic diabetes subtype.<br>• The m.3243A>G is the 4th most common genetic diagnosis in patients referred for MD testing.<br>• It must be tested in all patients with suspected MD, even if there is no hearing loss in the family, due to variable penetrance.<br>• It can be detected by NGS but requires specific targeting. Alternatively, a rapid quantitative genotyping assay such as pyrosequencing can be used.<br>• Sanger sequencing is possible but requires a minimum heteroplasmy detection level of ~5%. |
| GCK – Sanger & MLPA (Czech Republic) | 1 | 140 | 74% (103/140) | 1 | • The clinical phenotype is easily recognized in children and in pregnancy. |
| GCK – non-invasive prenatal testing by ddPCR (UK) | 1 | 33 | 100% concordance with cord blood result. | 3 | • This enables specific and rapid sequencing of GCK with a high diagnostic yield (>70%).<br>• A rapid diagnosis in pregnancy enables non-invasive prenatal testing to aid clinical management by using a digital PCR technique that is 100% accurate. |

established the etiological role of genes such as *HNF1A*, *HNF4A* and *GCK*, recent studies that have investigated variation in genes such as *BLK*, *KLF11* and *PAX4* in large population datasets have not supported their role in causing monogenic diabetes[73], and these genes were recently refuted as monogenic diabetes genes by the MDEP GCEP.

The consensus opinion of the writing group was that a gene should only be considered causative of monogenic diabetes if it meets the criteria set out in expertly curated guidelines that have been developed to validate gene-disease relationships. These guidelines have already been applied to many of the monogenic diabetes genes by the ClinGen MDEP GCEP. We recommend continued efforts to curate new and updated existing monogenic diabetes genes for gene-disease validity be centralized with the MDEP GCEP. Those interested in contributing to this effort should engage with the MDEP GCEP to ensure that genes used in monogenic diabetes have been curated for gene disease validity in a process that is evidence based and updated on a standard schedule as directed by ClinGen.

**Question 4—On what basis should a variant be considered a cause of monogenic diabetes?** In 2015, the ACMG and AMP developed general guidelines for the interpretation of sequence variants[38]. The ClinGen Sequence Variant Interpretation (SVI) Working Group has published multiple updates to these original guidelines[39–42]. The Association for Clinical Genomic Science (ACGS) voted to adopt these guidelines[74]. These guidelines have undergone several updates. ClinGen's MDEP VCEP has modified these general guidelines for three common monogenic diabetes genes (*HNF1A*, *HNF4A* and *GCK*); these guidelines account for many issues inherent in the difficulty in interpreting monogenic diabetes variants and can be used as a framework for interpreting variants in genes for which rules have not yet been established.

The ACMG/AMP guidelines were developed through an evidence-based process involving the sharing, developing, and validating of variant classification protocols among over 45 laboratories in North America. They incorporate various types of evidence to determine if a variant is pathogenic, likely pathogenic, of uncertain significance (VUS), likely benign, or benign. Examples of the types of evidence include: frequency in public databases such as gnomAD; the segregation of a variant with a disease phenotype; results of computational (in silico) prediction programs; de novo status; functional studies; frequency of variant in cases vs. controls; the presence of other pathogenic variants at the same nucleotide or within the same codon; the location of a variant (i.e., if it is within a well-established functional domain or mutational hotspot); and whether a variant has been found in a patient with a phenotype consistent with the disease. MDEP gene-specific rules incorporate experts' unpublished case data and knowledge of monogenic diabetes phenotype and prevalence in recommending the evidence and thresholds to apply.

Continued work by MDEP VCEP is needed to develop applications of the guidelines tailored to additional monogenic diabetes types and genes. Improvement in de-identified case-sharing platforms is needed to promote maximizing the ability to gather the evidence needed to evaluate pathogenicity.

**Question 5—How should a variant in a monogenic diabetes gene be reported?** Well written general guidelines for the reporting of genetic test results are available[43,44,75–79] and this review will therefore summarize the basic requirements and focus on reporting monogenic diabetes tests.

We summarize the recommendations for reporting results for a range of different testing scenarios and methodologies (Table 4). A single page report with appendices is preferred. The report

| Table 5 Recommendations based on synthesis of evidence for how to test for monogenic diabetes. | | |
|---|---|---|
| **Recommendation** | **How to test for monogenic diabetes** | **Grade** |
| 1 | A targeted NGS approach is the preferred option for testing for monogenic diabetes to maximize diagnostic yield without significant cost and variant interpretation burden compared to gene agnostic genome sequencing. Genome sequencing can provide data for novel gene and non-coding variant discovery and allows re-analysis for newly associated genes and variants but is prohibitively expensive for many laboratories and requires significant bioinformatics expertise to manage the huge numbers of variants and give correct classifications. | A |
| 2 | Targeted panels should be designed to include all known causes of monogenic diabetes including mitochondrial diabetes, detect known non-coding mutations (located in promoters, deep introns, and distal enhancers) and detect CNVs. A comprehensive gene panel that includes all recessively inherited genes is essential in countries and populations with high rates of consanguinity. | A |
| 3 | A separate MLPA assay for CNV detection or genotyping assay such as pyrosequencing for m.3243A>G detection is acceptable but comes at increased cost. | A |
| 4 | NDM testing services should offer a methylation-based assay such as MS-MLPA since 6q24 imprinting defects are a common cause of TNDM. | A |
| 5 | The high diagnostic yield for *GCK* in suspected GCK related hyperglycemia and for *KCNJ11*, *ABCC8* and *INS* in NDM diagnosed within 6-12 months, and the clinical utility of these diagnoses, justifies rapid Sanger sequencing of these genes initially in these scenarios. | A |

should restate the reason for testing, including the clinical characteristics/phenotype of the patient. The report must include a headline result or summary that clearly states the outcome of the test for the patient – this may be stating whether a diagnosis of monogenic diabetes has or has not been made, or whether a patient is or is not genetically predisposed to monogenic diabetes. Patients with specific subtypes may respond well to certain therapies and this should be noted in the report. Testing should be offered to at-risk family members, which may be diagnostic, predictive or carrier testing. Special care should be taken when reporting variants in syndromic diabetes genes in patients with isolated diabetes. The risk to future offspring should be stated according to mode of inheritance. The report should not use terms positive or negative for describing test results. Variants should be reported in a table that includes the HUGO gene name, zygosity of the variant, both nucleotide and protein level descriptions using HGVS nomenclature, genomic coordinates and the classification of the variant based on the ACMG/AMP 5 level classification system[38]. Benign and likely benign variants should not be reported. Class 3 (VUS or VOUS) variants should be reported based on professional judgment, the level of supporting evidence and on whether additional investigations can be undertaken to change the classification such as testing of other affected relatives, further biochemical testing, or additional functional laboratory investigations. Evidence used to classify the variant should be clearly outlined. Technical information should be provided in a section separate from results and interpretation and will include details of the methodology and gene or genes tested. If the testing performed does not cover all known genes and possible mutations, then this should be stated as a limitation with recommendations for further genetic testing (e.g., NGS or MLPA analysis). Laboratory reports should avoid the terminology "MODY" given its lack of precision in sensitivity or specificity for any given genetic etiology of monogenic diabetes. Instead the use of the gene name hyphenated with hyperglycemia, diabetes severe insulin resistance, lipodystrophy or syndrome as appropriate is recommended. Examples are GCK-related hyperglycemia, HNF1Adiabetes, INSR-severe insulin resistance, PPARG-lipodystrophy, mt.3243 A > G syndrome. Alternatively, the term "monogenic diabetes" followed by "subtype gene name" eg: monogenic diabetes subtype HNF4A may be used.

The structure, format, and content of monogenic diabetes testing reports will vary widely between laboratories across the world. Standardization is difficult due to variability in mandatory report content, such as legal disclaimers, and the ability to include clinical recommendation. But there are essential reporting best practices that should be adopted by all laboratories irrespective of local reporting policies. We recommend that laboratories performing monogenic diabetes testing participate in the EMQN's monogenic diabetes EQA scheme (www.emqn.org) which aims to educate and improve quality of diagnostic testing and reporting for this condition. Future research is advised to engage patients, providers, and other stakeholders in the design and evaluation of readability, comprehension, and application of information contained in genetic testing reports for monogenic diabetes.

**Question 6—Research Question: What are the next steps after a diagnosis of monogenic diabetes?** A systematic, comprehensive, and collaborative approach is required after making a monogenic diabetes diagnosis after conducting genetic testing. Our guidance for the next steps after diagnosis of monogenic diabetes focuses on the following: (1) practical recommendations for providing the diagnosis results and clinical follow-up, (2) reviewing genetic testing reports, (3) family testing for adults and children, (4) legal considerations for this diagnosis, (5) considering psychological impact of diagnosis, and (6) recommendations for addressing VUS results and negative monogenic diabetes testing despite atypical features to a patient's diabetes presentation. In the following paragraphs, the term "clinician" can refer to a physician or genetic counselor. Genetic counselors are specially trained to communicate complex genetic information, facilitate family testing, and address psychosocial issues that may arise with a new diagnosis; thus, we recommend having a genetic counselor as part of the care team if possible. Upon receipt of a genetic test result diagnosing monogenic diabetes (i.e., pathogenic, or likely pathogenic variant identified), the clinician should schedule a 30–60 min in-person or telehealth appointment with the patient/family[80]. We do not recommend that results be disclosed via an electronic health record (EHR) portal or by non-clinical staff.

After a very brief reminder of what the genetic test analyzed, we recommend the clinician describe the identified variant in patient-friendly language (e.g., a single spelling error in the genetic code) and review how disease-causing variants in the gene impair glucose metabolism. The clinician can explain the evidence used to classify the variant as disease-causing which is often included in the genetic testing report, e.g., if the variant was previously identified in patients with monogenic diabetes or

experimental evidence demonstrated loss of function. The clinician should describe the general features of the type of monogenic diabetes indicated by the genetic change, including the inheritance pattern of the disorder, specifying those features that are consistent with the patient's clinical picture. If the type of monogenic diabetes is characterized by variable expressivity and/or reduced penetrance, these concepts should be introduced to the patient/family, providing specific examples from the disorder at hand. HNF1B syndrome is a prime example of variable expressivity, as the renal and extra-renal phenotypes (diabetes, genital malformations, pancreatic hypoplasia, abnormal liver function) vary among affected individuals, even within the same family[81,82]. The patient/family should be provided a copy of the report for their records. Additionally, a document describing the variant identified and avenues for variant-specific testing can be provided to the patient to distribute to family members if family testing is being pursued. Upon reflecting on the diagnosis, patients may feel relief at a genetic etiology for their symptoms, while others may feel angry or annoyed if they were initially misdiagnosed and prescribed suboptimal treatment[83–88]. Feelings of frustration should be validated. Some patients may find solace in hearing that knowledge and testing of monogenic diabetes have both evolved greatly over time and we hope more diagnoses will be made moving forward. Patients may also be helped by speaking to other patients with monogenic diabetes. At this time, formal support groups are limited for monogenic diabetes, but the provider can consider connecting patients with monogenic diabetes given mutual consent. Patients, providers, and researchers are in the process of creating a consortium for communication and support regarding monogenic diabetes called the Monogenic Diabetes Research and Advocacy Consortium (MDRAC, mdrac.org). Yearly follow-up can be suggested to continue to provide updates on the monogenic diabetes diagnosis, prognosis, and treatment in addition to any new information on the gene and genetic variant identified.

Results of genetic testing should be discussed in context of the family history. The most common forms of monogenic diabetes, HNF1A, HNF4A, and GCK etiologies, are dominantly inherited, and the vertical transmission of diabetes or hyperglycemia is often evident in the pedigree[8,89–91]. If a disease-causing variant in one of these conditions is identified in a parent of an affected individual, there is a 50% chance that siblings and children of the proband will inherit the variant. The absence of a family history of diabetes may suggest that a variant associated with a dominant condition is de novo in the proband. If parents test negative and maternity and paternity are confirmed, the recurrence risk in siblings is approximately 1%, which accounts for the possibility of gonadal mosaicism[92]. De novo disease-causing variants have been reported and are especially common in HNF1B[81,93,94]. With HNF1B etiology, the family history may also include genital tract malformations, renal cysts, or pancreatic hypoplasia[81]. Recurrence risk of recessive forms of monogenic diabetes, such as Wolcott-Rallison syndrome (due to EIF2AK3 mutations) or Thymine Responsive Megaloblastic Anemia (TRMA) syndrome, is 25% in offspring when both the proband and their partner are carriers[89]. Recurrence risk of monogenic diabetes caused by the mitochondrial DNA MIDD (maternally inherited diabetes and deafness) variant (m.3243A>G) is essentially zero when the sperm-producing parent has the variant, as mitochondria are passed down through the oocyte. All offspring and maternal relatives of the egg-producing parent will inherit the variant, albeit at varying heteroplasmy[95].

Affected family members of individuals with molecular confirmation of monogenic diabetes should be offered variant-specific testing of the familial variant, a process known as cascade testing[8,80]. For probands with GCK- related hyperglycemia, it is important to also discuss cascade testing of family members with gestational diabetes and pre-diabetes, since this is characterized by stable, mild fasting hyperglycemia that is clinically asymptomatic and can also impact pregnancy management in a gestational parent with apparent GDM. Aparently unaffected or undiagnosed first-degree adult relatives of probands with GCK etiology should be counseled that if they wish to undergo a fasting glucose test; if normal, a diagnosis of GCK- hyperglycemia is highly unlikely and genetic testing is unnecessary[8,96,97]. If they have an elevated fasting glucose test, then they should undergo a cascade genetic test to clarify whether GCK is the etiology.

The risks and benefits of testing, and possible results of testing, should be reviewed in all cases to allow the family to make autonomous testing decisions consistent with their goals and values. Possible benefits of genetic testing include the ability to obtain or advocate for more appropriate treatment, reduced anxiety, and uncertainty, decreased stigma, knowledge of recurrence risk, and the ability to plan for the future[80,83,87,98]. Risks may include increased anxiety, trouble adjusting to a new diagnosis, or learning unexpected information[85]. The risk of insurance discrimination may also need to be reviewed, as different countries have instituted varying rules regulating the use of genetic information in insurance underwriting.

Additional ethical and psychosocial issues surrounding a genetic diagnosis should also be discussed when considering predictive testing in a minor. The clinical relevance of an HNF1A or HNF4A positive genetic test would likely have minimal clinical relevance prior to adolescence, and thus we generally discourage genetic testing in young children. Indeed, adolescents in families with HNF1A diabetes preferred testing in adolescence when parents and their children can engage in joint decision-making regarding genetic testing[84]. We do not recommend testing asymptomatic children for GCK, given this is a benign condition and there are potential adverse psychosocial effects of being labeled as "sick"[98]. Also, the "GCK related hyperglycemia" diagnosis can lead to problems achieving life, long-term care, or disability insurances since this may be classified as a monogenic form of diabetes. However, in practice this is a benign condition that often fulfills the glucose criteria for prediabetes rather than diabetes and does not impact any increased cardiovascular risk or progression to type 2 diabetes that is frequently seen in people with prediabetes. If a child in a family with a GCK etiology is incidentally found to have hyperglycemia, their pediatrician should be informed of the familial variant and familial variant testing can proceed to avoid unnecessary treatment.

These strategies also respect the autonomy of the child to make an informed decision about testing when they are able. However, the significant fear of uncertainty that some parents of at-risk children feel should not be dismissed. Genetic testing may decrease anxiety in parents, allow them to gradually introduce the disorder to their child in a developmentally appropriate manner, and empower them to prepare for the future[83,98].

A positive result of genetic testing would replace the prior diagnosis (of type 1 or type 2 diabetes) with a diagnosis of monogenic diabetes. The clinician should review the prognosis of the condition and potential changes in medical management (e.g., no treatment in GCK- hyperglycemia and sulfonylurea treatment with HNF1A and HNF4A monogenic diabetes[8,97]. We refer the reader to the recommendations generated by the Monogenic Diabetes Precision Prognostics and Therapeutics groups for additional information which we also provide a high-level summary of in Fig. 2.

There may be instances when an asymptomatic family member has a positive result on genetic testing. In this case, the high risk of developing diabetes or hyperglycemia should be emphasized and a plan for monitoring blood glucose should be developed if

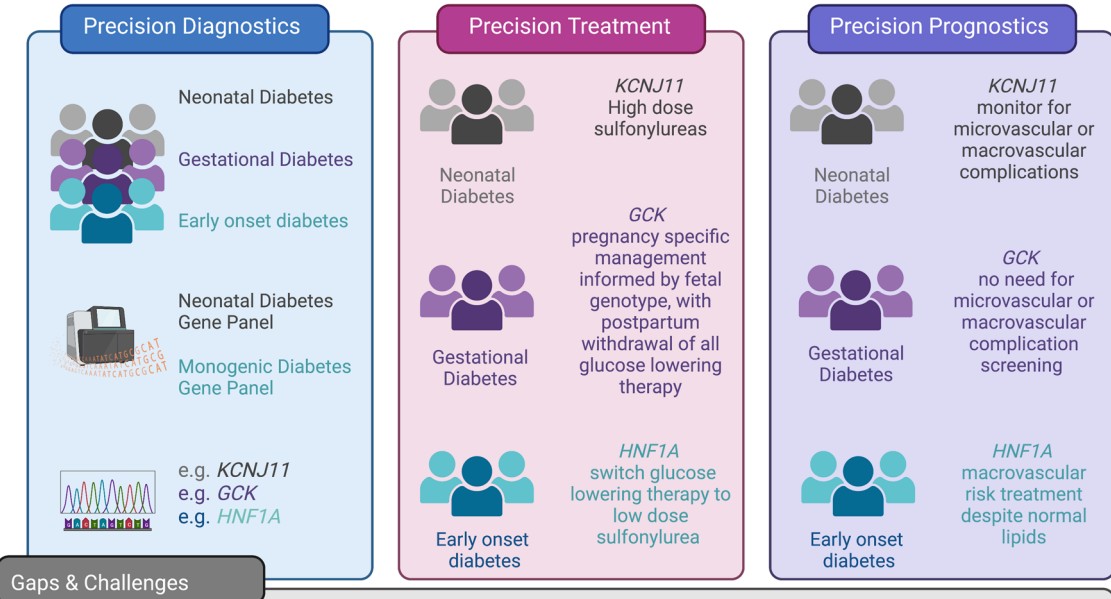

**Fig. 2 Schematic overview of how precision diagnostics leads to precision treatment and precision prognostics.** Examples of genetic forms of diabetes identified through precision diagnostics and how these lead to precision treatment and prognostics. Current gaps and challenges identified through the systematic review are highlighted.

appropriate. Of course, an asymptomatic family member with negative variant-specific testing may still be at risk of developing more common forms of diabetes[80]. A notable exception to this is that asymptomatic family members, with negative genetic testing of probands with MIDD, are still at risk for diabetes, hearing loss, and potentially other symptoms of mitochondrial disease given the variability in heteroplasmy of the m.3243A>G variant among different body tissues[95]. A negative result of variant-specific testing in a family member with diabetes would indicate another etiology for the diabetes diagnosis, such as type 1 or type 2 diabetes.

**Question 7—What are the current challenges for the field in precision diagnostics for monogenic diabetes?** We reviewed the abstracts of 455 abstracts and the full text of 42 papers before extracting data from 14 articles meeting our criteria. Several key themes emerged which present on-going challenges for the field of precision diagnostics for monogenic diabetes. With the generation of exome and now genome sequencing data from both larger clinical cohorts and biobanks we have new insights into variant frequency and penetrance. For some previously reported pathogenic variants in monogenic diabetes genes there is now evidence for reduced penetrance in unselected populations and that individuals carrying the variants do not necessarily display the hallmark characteristics (e.g BMI < 30 kg/m$^2$, age of diagnosis <35 years) of monogenic diabetes[99–102]. The interpretation of novel rare variants in monogenic diabetes genes is challenging, functional studies can assist but multiple assays are required in concert with frequency and clinical data[103,104]. Functional studies are slow, lack standardization and are usually retrospectively performed after variant discovery. Efforts to generate variant maps in genes of interest are a potential route forward but will require coordination and implementation of standards[105]. The perpetuation of errors in the literature remains a concern with ongoing reporting of novel variants in genes which are not considered by experts in the field to be causal for monogenic

diabetes[106]. Whilst the reporting of potential novel genes can be misleading as they do not necessarily meet the criteria for classification as a novel genetic cause of diabetes[107]. There remain inequalities in sequencing data across diverse ancestries and populations even when there are examples of the importance of rare variation in monogenic diabetes genes[108–110].

Finally, barriers to genetic testing remain including limited provider awareness of monogenic diabetes. It is important that all clinicians treating diabetes patients are considering monogenic etiologies as a potential diagnosis, especially when diagnosis can occur in adults that have had diabetes since youth[111]. Future research should focus on increasing representation of sequence data in monogenic diabetes genes in diverse populations, generating variant maps of clinically actionable diabetes genes and continued efforts to share knowledge and expertise of monogenic diabetes in underserved communities and populations.

## Discussion
To diagnose monogenic diabetes offers an opportunity to find those who can benefit from precision medicine[89,112]. This systematic review has summarized and quantified data on the yield of monogenic diabetes detection using various selection criteria in different populations with diabetes. The greatest yield for monogenic diabetes is in those diagnosed with diabetes in the first 6 months of age, when the background prevalence of type 1 diabetes is low. There is progressively lower additional yield in those diagnosed between 6–9 months, or 6–12 months or above 12 months. Dominantly inherited subtypes such as KCNJ11, ABCC8 and INS account for the majority of neonatal diabetes in non-consanguineous families, while in consanguineous populations recessively inherited subtypes such as EIF2AK3 are more common. The ongoing detection of the commonest subtype of neonatal diabetes due to *KCNJ11/ABCC8* mutations who can be transferred to inexpensive sulfonylurea treatment makes testing all those below 12 months still cost-saving, so long as 3% of those

screened have a monogenic diabetes diagnosis that is treatment-changing[8]. Further large-scale studies need to be performed using large-gene panels in those with diabetes diagnosed between 12 and 24 months to ascertain further triaging features that could improve the yield of diagnosis of monogenic diabetes cases amongst the increasing majority who would have type 1 diabetes during early childhood. The prevalence of type 2 diabetes is generally only seen with severe obesity in early childhood with onset generally well above 24 months of age, so excluding obesity in those selected for genetic testing is less of a consideration in this very low age group. With increasing age of diabetes diagnosis, there are additional criteria required to lower the probability of type 1 diabetes (such as autoantibody negative and or retained C-peptide), and type 2 diabetes (such as the absence of obesity).

The recommendations for restricting testing to the selected groups of individuals with diabetes provided in this systematic review, have considered the highest diagnostic yields reported in the literature, balancing high specificity of clinical and/or biochemical features with high sensitivity for monogenic diabetes. As sensitivity for identifying monogenic diabetes subtypes increases with the use of less stringent clinical characteristics, the number needed to test to find one positive case increases, so choosing absolute thresholds of key characteristics is a matter of balancing the costs of the genetic test, against the benefits of correctly identifying individuals with monogenic diabetes. Different health systems should consider adapting these recommended thresholds for their population contexts to allow for systematic genetic testing in those with diabetes whose probability of having a monogenic diabetes etiology meets their cost-effectiveness threshold for implementing through clinical care pathways for diabetes.

Currently, we cannot afford to offer a genetic test to everyone with diabetes. Such an "all testing" approach would find every case of monogenic diabetes but would have a low diagnostic yield. In a USA study, monogenic diabetes genetic screening approach was modeled as being cost-effective at 6% yield and cost-saving at 30% yield for GCK, HNF1A, HNF4A, with an estimated combined prevalence of 2%, implicating the most clinically actionable changes to therapy[113]. In a UK study, selecting individuals for genetic testing among those diagnosed below 30 years, using either a clinical prediction model or a biomarker strategy (with negative antibodies and retained C-peptide) was deemed cost-saving, assuming cost-benefit from stopping insulin treatment for misdiagnosed type 1 diabetes in this age group[114]. This model assumed a prevalence of the most cost-saving monogenic diabetes types as 2.4% (GCK 0.7%, HNF1A 1.5% and HNF4A 0.2%). Clearly, once monogenic diabetes has been identified in a proband, cascade screening of family members maximizes cost-savings, given the yield is at least 50% in autosomal dominant subtypes of monogenic diabetes.

The use of NGS targeted large-gene panel is the recommended testing technology with variant curation developed by ClinGen. However, the high diagnostic yield of GCK etiology in patients with persistently mild fasting hyperglycemia means Sanger sequencing of this gene alone may be offered for such individuals. Laboratory reports should replace the imprecise term MODY with monogenic diabetes and the gene name.

In this article, we provide recommendations on practical steps for communicating a diagnosis of monogenic diabetes to patients, methods for family testing, and considering the psychological impact of diagnosis (Table 3 & 5; Fig. 2). The practice of communicating genetic testing results for monogenic diabetes to patients with a genetic diagnosis is evolving as monogenic diabetes testing becomes more prevalent. Although communicating genetic testing results for disease-causing variants is more straightforward, it remains challenging in communicating results of a VUS or a no genetic diagnosis resulting in a patient with a clearly atypical presentation of diabetes. Fortunately, collaborative efforts in variant curation and precision medicine research will continue to reduce the ambiguity in VUS or no diagnosis results and improve our ability to effectively provide recommendations for diagnosis, treatment and family testing for people who undergo monogenic diabetes testing.

The major strength with our study is that to the best of our knowledge it is the first comprehensive overview of all available evidence on diagnostics of monogenic diabetes based on screening more than 12,500 peer reviewed articles published during the last 32 years extracting data from >100 studies that met the predefined criteria. This makes it easier for healthcare professionals to make evidence-based decisions. We used rigorous and transparent methods, leading to a higher quality of the evidence for how to use precision diagnostics in monogenic diabetes than other types of studies. Moreover, we aimed to reduce bias in the selection of studies, data extraction, and analysis making our findings more reliable and credible. Finally, our systematic review is an efficient way to identify knowledge gaps and prioritize future research, as it avoids duplication of efforts and resources.

Our study also has several limitations. For the question of who to test, the index/triage test of clinical or laboratory biomarkers used to select people for monogenic diabetes testing would ideally be compared with the reference standard of genetically testing all individuals with diabetes (without any such selection) however, such studies were rare. Most were cohort or cross-sectional studies in patients with diabetes diagnostic uncertainty, that only genetically tested a smaller sample by certain criteria, so it was not possible to discern the number of cases missed (false negatives) with this approach. Only a few studies directly compared two or more approaches in the same study population, so most recommendations were based on comparative yields in different populations. Whilst syndromic forms of monogenic diabetes (such as mitochondrial diabetes and deafness, severe insulin resistance, lipodystrophy) were included in the search strategy, there were not sufficient papers that included at least 100 genetically tested individuals to permit graded recommendations on whom to select for genetic testing when these additional features were present.

Our review was limited by the availability of relevant studies, and sometimes there were not enough high-quality studies to draw meaningful conclusions. Hence, we were not able to address Questions 3-7 initially or ultimately (for Questions 6-7) by a systematic review using the method offered by Covidence. However, the co-authors have been working on diagnostics of monogenic diabetes for 10-30 years and are experts in the field. We therefore used expert opinion for the Questions 3–7. Another weakness is that only papers in the English language were included in the analysis. Thus, non-English papers potentially offering useful information on Questions 1-2 were missed. It is, however, not likely since we defined a cut-off of 100 study individuals undergoing genetic testing to ensure a high scientific quality. Conducting the systematic review was a time-consuming process searching for and evaluating many studies. It was also resource-extensive necessitating a team of trained researchers and specialized software. And we cannot completely exclude publication bias, where only studies with significant results are published, and non-significant results are not reported. Despite some limitations, we believe our systematic review will prove a valuable tool in precision diagnostics of monogenic diabetes providing high-quality evidence to inform clinical decision-making.

What is needed next? Our systematic review reveals that improved access to genetic testing for monogenic diabetes to prevent health disparities is important. There are issues regarding equity and utility in non-European countries where background prevalence of type 2 diabetes is higher. Ethnicity dependent thresholds for overweight or obese categories need to be considered

when implementing the recommendations of whom to test in different geographical regions with distinct ethnic groups. Cultural factors may also influence the acceptability of screening for monogenic diabetes which needs further study and education. Moreover, type 1 diabetes genetic risk score (a tool for using common susceptibility variants for type 1 diabetes to pre-assess the likelihood of having type 1 vs. other types of diabetes) data has not been well characterized in these countries. Another step is generation of and access to systematic measurements of autoantibodies and C-peptide for people diagnosed with diabetes under the age of 45 years with the addition of validated ancestry-appropriate type 1 diabetes genetic risk score data. This information would be advantageous to better discriminate monogenic diabetes from type 1 diabetes. What is more, improvement in de-identified case-sharing platforms is needed to promote maximizing the ability to gather the evidence needed to evaluate pathogenicity. As such, continued work by expert panels such as the MDEP VCEP is warranted to develop guidelines for which gene variants should be considered causative of monogenic diabetes as well as applications of the guidelines tailored to additional monogenic diabetes types and genes. One relevant instrument is generation of deep mutational scanning maps of monogenic diabetes genes to aid variant classification. It is also important to remember that the genetic and genomic testing landscape is ever evolving, with a strong possibility of universal genome sequencing in the future, which would reduce concerns on whom to test but place an even higher burden on having adequate tools, expertise, and workforce for interpretation. Finally, with the increased numbers of people being given a genetic diagnosis of monogenic diabetes, further studies to evaluate whether this actually results in improved management due to capacity of medical services are needed. Further clinical guidance is needed for steps following monogenic diabetes testing which includes genetic counseling, subsequent referrals, and family testing in addition to research on the outcomes of implementation.

## Data availability

Complete lists of the publications where data were extracted for this study are provided in Supplementary Data 1 and 2.

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

## Acknowledgements

The ADA/EASD Precision Diabetes Medicine Initiative (PMDI), within which this work was conducted, has received the following support: The Covidence license was funded by Lund University (Sweden) for which technical support was provided by Maria Björklund and Krister Aronsson (Faculty of Medicine Library, Lund University, Sweden). Administrative support was provided by Lund University (Malmö, Sweden), University of Chicago (IL, USA), and the American Diabetes Association (Washington D.C., USA). The Novo Nordisk Foundation (Hellerup, Denmark) provided grant support for in-person writing group meetings (PI: L Phillipson, University of Chicago, IL).

S.M. has a personal award from Wellcome Trust Career Development scheme (223024/Z/21/Z) and holds Institutional funds from the NIHR Biomedical Research Centre Funding Scheme. P.R.N. was supported by grants from the European Research Council (AdG #293574), Stiftelsen Trond Mohn Foundation (Mohn Center of Diabetes Precision Medicine), the University of Bergen, Haukeland University Hospital, the Research Council of Norway (FRIPRO grant #240413), the Western Norway Regional Health Authority (Strategic Fund "Personalised Medicine for Children and Adults"), and the Novo Nordisk Foundation (grant #54741). J.M.I. was supported by a grant from the National Institute of Health (K08DK133676). S.E.F. is supported by a Wellcome Trust Senior Research Fellowship (Grant Number 223187/Z/21/Z). ALG is a Wellcome Trust Senior Fellow (200837/Z/16/Z) and is also supported by NIDDK (UM-1DK126185). EDF is a Diabetes UK RD Lawrence Fellow (19/005971). KRO is supported by the National Institute for Health Research (NIHR) Oxford Biomedical Research Centre (BRC). The views expressed are those of the author(s) and not necessarily those of the NHS, the NIHR or the Department of Health.

## Author contributions

Review design: R.M., K.C., T.I.P., P.R.N., L.K.B., K.R.O., A.L.G. Systematic review implementation: R.M., K.C., T.I.P., J.I.M., P.S., K.A.M., C.S., J.M., S.M., I.A., E.dF., S.E.F., P.R.N., L.K.B., K.R.O., A.L.G. Data extraction manuscripts: R.M., K.C., T.I.P., J.M.I., P.S., L.K.B., K.R.O., A.L.G. Manuscript writing: R.M., K.C., T.I.P., L.K.B., K.A.M., P.R.N., K.R.O., A.L.G. Project Management: R.M., L.K.B., K.R.O., A.L.G.

## Competing interests

R.M. has received honoraria for consulting or educational activities for Eli Lilly, Novo Nordisk and Boeringer Ingelheim. S.M. has Investigator initiated funding from DexCom and serves on the Board of Trustees for the Diabetes Research & Wellness Foundation (UK). LK Billings: Received honoraria for consulting activities for Novo Nordisk, Bayer, Lilly, Xeris, and Sanofi which are unrelated to this present work. ALG's spouse holds stock options in Roche and is an employee of Genentech. No other authors have competing interests.

## Additional information

[1]Department of Medicine, Faculty of Medical and Health Sciences, University of Auckland, Auckland, New Zealand. [2]Auckland Diabetes Centre, Te Whatu Ora Health New Zealand, Te Tokai Tumai, Auckland, New Zealand. [3]Exeter Genomics Laboratory, Royal Devon University Healthcare NHS Foundation Trust, Exeter, United Kingdom. [4]Department of Medicine, University of Maryland School of Medicine, Baltimore, Maryland, USA. [5]Department of Pediatrics, Division of Endocrinology & Diabetes, Stanford School of Medicine, Stanford, CA, USA. [6]Stanford Diabetes Research Center, Stanford School of Medicine, Stanford, CA, USA. [7]Children and Youth Clinic, Haukeland University Hospital, Bergen, Norway. [8]Mohn Center for Diabetes Precision Medicine, Department of Clinical Science, University of Bergen, Bergen, Norway. [9]Department of Medical Genetics, AP-HP Pitié-Salpêtrière Hospital, Sorbonne University, Paris, France. [10]Department of Medical Genetics, Haukeland University Hospital, Bergen, Norway. [11]Department of Metabolism, Digestion and Reproduction, Imperial College London, London, UK. [12]Department of Diabetes and Endocrinology, Imperial College Healthcare NHS Trust, London, UK. [13]Department of Clinical and Biomedical Science, Faculty of Health and Life Sciences, University of Exeter, Exeter, UK. [14]Division of Endocrinology, NorthShore University HealthSystem, Skokie, IL, USA. [15]Department of Medicine, Pritzker School of Medicine, University of Chicago, Chicago, IL, USA. [16]Oxford Center for Diabetes, Endocrinology & Metabolism, University of Oxford, Oxford, UK. [17]NIHR Oxford Biomedical Research Centre, Oxford, UK. [18]Department of Genetics, Stanford School of Medicine, Stanford, CA, USA. [208]These authors contributed equally: Rinki Murphy, Kevin Colclough, Toni I. Pollin. [209]These authors jointly supervised this work: Liana K. Billings, Katharine R Owen, Anna L Gloyn. *A list of authors and their affiliations appears at the end of the paper.
✉email: R.murphy@auckland.ac.nz; agloyn@stanford.edu

## ADA/EASD PMDI

Deirdre K. Tobias[19,20], Jordi Merino[21,22,23], Abrar Ahmad[24], Catherine Aiken[25,26], Jamie L. Benham[27], Dhanasekaran Bodhini[28], Amy L. Clark[29], Kevin Colclough[30], Rosa Corcoy[31,32,33], Sara J. Cromer[22,34,35], Daisy Duan[36], Jamie L. Felton[37,38,39], Ellen C. Francis[40], Pieter Gillard[41], Véronique Gingras[42,43], Romy Gaillard[44], Eram Haider[45], Alice Hughes[30], Jennifer M. Ikle[46,47], Laura M. Jacobsen[48], Anna R. Kahkoska[49], Jarno L. T. Kettunen[50,51,52], Raymond J. Kreienkamp[22,23,34,53], Lee-Ling Lim[54,55,56], Jonna M. E. Männistö[57,58], Robert Massey[45], Niamh-Maire Mclennan[59], Rachel G. Miller[60], Mario Luca Morieri[61,62], Jasper Most[63], Rochelle N. Naylor[64], Bige Ozkan[65,66], Kashyap Amratlal Patel[30], Scott J. Pilla[67,68], Katsiaryna Prystupa[69,70], Sridaran Raghaven[71,72], Mary R. Rooney[65,73], Martin Schön[69,70,74], Zhila Semnani-Azad[20], Magdalena Sevilla-Gonzalez[34,35,75], Pernille Svalastoga[7,8], Wubet Worku Takele[76], Claudia Ha-ting Tam[56,77,78], Anne Cathrine B. Thuesen[21], Mustafa Tosur[79,80,81], Amelia S. Wallace[65,73], Caroline C. Wang[73], Jessie J. Wong[82], Jennifer M. Yamamoto[83], Katherine Young[30], Chloé Amouyal[84,85], Mette K. Andersen[21], Maxine P. Bonham[86], Mingling Chen[87], Feifei Cheng[88], Tinashe Chikowore[35,89,90,91], Sian C. Chivers[92], Christoffer Clemmensen[21], Dana Dabelea[93], Adem Y. Dawed[45], Aaron J. Deutsch[23,34,35], Laura T. Dickens[94], Linda A. DiMeglio[37,38,39,95], Monika Dudenhöffer-Pfeifer[24], Carmella Evans-Molina[37,38,39,96], María Mercè Fernández-Balsells[97,98], Hugo Fitipaldi[24], Stephanie L. Fitzpatrick[99], Stephen E. Gitelman[100], Mark O. Goodarzi[101,102], Jessica A. Grieger[103,104], Marta Guasch-Ferré[20,105], Nahal Habibi[103,104], Torben Hansen[21], Chuiguo Huang[56,77], Arianna Harris-Kawano[37,38,39], Heba M. Ismail[37,38,39], Benjamin Hoag[106,107], Randi K. Johnson[108,109], Angus G. Jones[30,110], Robert W. Koivula[111], Aaron Leong[22,35,112], Gloria K. W. Leung[86], Ingrid M. Libman[113], Kai Liu[103], S. Alice Long[114], William L. Lowe Jr.[115], Robert W. Morton[116,117,118], Ayesha A. Motala[119], Suna Onengut-Gumuscu[120], James S. Pankow[121], Maleesa Pathirana[103,104], Sofia Pazmino[122], Dianna Perez[37,38,39], John R. Petrie[123], Camille E. Powe[22,34,35,124], Alejandra Quinteros[103], Rashmi Jain[125,126], Debashree Ray[73,127], Mathias Ried-Larsen[128,129], Zeb Saeed[130], Vanessa Santhakumar[19], Sarah Kanbour[67,131], Sudipa Sarkar[67], Gabriela S. F. Monaco[37,38,39], Denise M. Scholtens[132], Elizabeth Selvin[65,73], Wayne Huey-Herng Sheu[133,134,135], Cate Speake[136], Maggie A. Stanislawski[108], Nele Steenackers[122], Andrea K. Steck[137], Norbert Stefan[70,138,139], Julie Støy[140], Rachael Taylor[141], Sok Cin Tye[142,143], Gebresilasea Gendisha Ukke[76], Marzhan Urazbayeva[80,144], Bart Van der Schueren[122,145], Camille Vatier[146,147], John M. Wentworth[148,149,150], Wesley Hannah[151,152], Sara L. White[92,153], Gechang Yu[56,77], Yingchai Zhang[56,77], Shao J. Zhou[104,154], Jacques Beltrand[155,156], Michel Polak[155,156], Ingvild Aukrust[8,10], Elisa de Franco[30], Sarah E. Flanagan[30], Kristin A. Maloney[157], Andrew McGovern[30], Janne Molnes[8,10], Mariam Nakabuye[21], Pål Rasmus Njølstad[7,8], Hugo Pomares-Millan[24,158], Michele Provenzano[159], Cécile Saint-Martin[9], Cuilin Zhang[160,161], Yeyi Zhu[162,163],

Sungyoung Auh[164], Russell de Souza[117,165], Andrea J. Fawcett[166,167], Chandra Gruber[168], Eskedar Getie Mekonnen[169,170], Emily Mixter[171], Diana Sherifali[117,172], Robert H. Eckel[173], John J. Nolan[174,175], Louis H. Philipson[171], Rebecca J. Brown[164], Liana K. Billings[14,176], Kristen Boyle[93], Tina Costacou[60], John M. Dennis[30], Jose C. Florez[22,23,34,35], Anna L. Gloyn[46,47,177], Maria F. Gomez[24,178], Peter A. Gottlieb[137], Siri Atma W. Greeley[179], Kurt Griffin[126,180], Andrew T. Hattersley[30,110], Irl B. Hirsch[181], Marie-France Hivert[22,182,183], Korey K. Hood[82], Jami L. Josefson[166], Soo Heon Kwak[184], Lori M. Laffel[185], Siew S. Lim[76], Ruth J. F. Loos[21,186], Ronald C. W. Ma[56,77,78], Chantal Mathieu[41], Nestoras Mathioudakis[67], James B. Meigs[35,112,187], Shivani Misra[188,189], Viswanathan Mohan[190], Rinki Murphy[191,192,193], Richard Oram[30,110], Katharine R. Owen[111,194], Susan E. Ozanne[195], Ewan R. Pearson[45], Wei Perng[93], Toni I. Pollin[157,196], Rodica Pop-Busui[197], Richard E. Pratley[198], Leanne M. Redman[199], Maria J. Redondo[79,80], Rebecca M. Reynolds[59], Robert K. Semple[59,200], Jennifer L. Sherr[201], Emily K. Sims[37,38,39], Arianne Sweeting[202,203], Tiinamaija Tuomi[50,149,52], Miriam S. Udler[22,23,34,35], Kimberly K. Vesco[204], Tina Vilsbøll[205,206], Robert Wagner[69,70,207], Stephen S. Rich[120] & Paul W. Franks[20,24,111,118]

[19]Division of Preventative Medicine, Department of Medicine, Brigham and Women's Hospital and Harvard Medical School, Boston, MA, USA. [20]Department of Nutrition, Harvard T.H. Chan School of Public Health, Boston, MA, USA. [21]Novo Nordisk Foundation Center for Basic Metabolic Research, Faculty of Health and Medical Sciences, University of Copenhagen, Copenhagen, Denmark. [22]Diabetes Unit, Endocrine Division, Massachusetts General Hospital, Boston, MA, USA. [23]Center for Genomic Medicine, Massachusetts General Hospital, Boston, MA, USA. [24]Department of Clinical Sciences, Lund University Diabetes Centre, Lund University Malmö, Sweden. [25]Department of Obstetrics and Gynaecology, the Rosie Hospital, Cambridge, UK. [26]NIHR Cambridge Biomedical Research Centre, University of Cambridge, Cambridge, UK. [27]Departments of Medicine and Community Health Sciences, Cumming School of Medicine, University of Calgary, Calgary, AB, Canada. [28]Department of Molecular Genetics, Madras Diabetes Research Foundation, Chennai, India. [29]Division of Pediatric Endocrinology, Department of Pediatrics, Saint Louis University School of Medicine, SSM Health Cardinal Glennon Children's Hospital, St. Louis, MO, USA. [30]Department of Clinical and Biomedical Sciences, University of Exeter Medical School, ExeterDevonUK. [31]CIBER-BBN, ISCIII, Madrid, Spain. [32]Institut d'Investigació Biomèdica Sant Pau (IIB SANT PAU), Barcelona, Spain. [33]Departament de Medicina, Universitat Autònoma de Barcelona, Bellaterra, Spain. [34]Programs in Metabolism and Medical & Population Genetics, Broad Institute, Cambridge, MA, USA. [35]Department of Medicine, Harvard Medical School, Boston, MA, USA. [36]Division of Endocrinology, Diabetes and Metabolism, Johns Hopkins University School of Medicine, Baltimore, MD, USA. [37]Department of Pediatrics, Indiana University School of Medicine, Indianapolis, IN, USA. [38]Herman B Wells Center for Pediatric Research, Indiana University School of Medicine, Indianapolis, IN, USA. [39]Center for Diabetes and Metabolic Diseases, Indiana University School of Medicine, Indianapolis, IN, USA. [40]Department of Biostatistics and Epidemiology, Rutgers School of Public Health, Piscataway, NJ, USA. [41]University Hospital Leuven, Leuven, Belgium. [42]Department of Nutrition, Université de Montréal, Montreal, Quebec, Canada. [43]Research Center, Sainte-Justine University Hospital Center, Montreal, Quebec, Canada. [44]Department of Pediatrics, Erasmus Medical Center, Rotterdam, The Netherlands. [45]Division of Population Health & Genomics, School of Medicine, University of Dundee, Dundee, UK. [46]Department of Pediatrics, Stanford School of Medicine, Stanford University, Stanford, CA, USA. [47]Stanford Diabetes Research Center, Stanford School of Medicine, Stanford University, Stanford, CA, USA. [48]University of Florida, Gainesville, FL, USA. [49]Department of Nutrition, University of North Carolina at Chapel Hill, Chapel Hill, NC, USA. [50]Helsinki University Hospital, Abdominal Centre/Endocrinology, Helsinki, Finland. [51]Folkhalsan Research Center, Helsinki, Finland. [52]Institute for Molecular Medicine Finland FIMM, University of Helsinki, Helsinki, Finland. [53]Department of Pediatrics, Division of Endocrinology, Boston Children's Hospital, Boston, MA, USA. [54]Department of Medicine, Faculty of Medicine, University of Malaya, Kuala Lumpur, Malaysia. [55]Asia Diabetes Foundation, Hong Kong SAR, China. [56]Department of Medicine & Therapeutics, Chinese University of Hong Kong, Hong Kong SAR, China. [57]Departments of Pediatrics and Clinical Genetics, Kuopio University Hospital, Kuopio, Finland. [58]Department of Medicine, University of Eastern Finland, Kuopio, Finland. [59]Centre for Cardiovascular Science, Queen's Medical Research Institute, University of Edinburgh, Edinburgh, UK. [60]Department of Epidemiology, University of Pittsburgh, Pittsburgh, PA, USA. [61]Metabolic Disease Unit, University Hospital of Padova, Padova, Italy. [62]Department of Medicine, University of Padova, Padova, Italy. [63]Department of Orthopedics, Zuyderland Medical Center, Sittard-Geleen, The Netherlands. [64]Departments of Pediatrics and Medicine, University of Chicago, Chicago, Illinois, USA. [65]Welch Center for Prevention, Epidemiology, and Clinical Research, Johns Hopkins Bloomberg School of Public Health, Baltimore, Maryland, USA. [66]Ciccarone Center for the Prevention of Cardiovascular Disease, Johns Hopkins School of Medicine, Baltimore, MD, USA. [67]Department of Medicine, Johns Hopkins University, Baltimore, MD, USA. [68]Department of Health Policy and Management, Johns Hopkins University Bloomberg School of Public Health, Baltimore, Maryland, USA. [69]Institute for Clinical Diabetology, German Diabetes Center, Leibniz Center for Diabetes Research at Heinrich Heine University Düsseldorf, Auf'm Hennekamp 65, 40225 Düsseldorf, Germany. [70]German Center for Diabetes Research (DZD), Ingolstädter Landstraße 1, 85764 Neuherberg, Germany. [71]Section of Academic Primary Care, US Department of Veterans Affairs Eastern Colorado Health Care System, Aurora, CO, USA. [72]Department of Medicine, University of Colorado School of Medicine, Aurora, CO, USA. [73]Department of Epidemiology, Johns Hopkins Bloomberg School of Public Health, Baltimore, Maryland, USA. [74]Institute of Experimental Endocrinology, Biomedical Research Center, Slovak Academy of Sciences, Bratislava, Slovakia. [75]Clinical and Translational Epidemiology Unit, Massachusetts General Hospital, Boston, MA, USA. [76]Eastern Health Clinical School, Monash University, Melbourne, Victoria, Australia. [77]Laboratory for Molecular Epidemiology in Diabetes, Li Ka Shing Institute of Health Sciences, The Chinese University of Hong Kong, Hong Kong, China. [78]Hong Kong Institute of Diabetes and Obesity, The Chinese University of Hong Kong, Hong Kong, China. [79]Department of Pediatrics, Baylor College of Medicine, Houston, TX, USA. [80]Division of Pediatric Diabetes and Endocrinology, Texas Children's Hospital, Houston, TX, USA. [81]Children's Nutrition Research Center, USDA/ARS, Houston, TX, USA. [82]Stanford University School of Medicine, Stanford, CA, USA. [83]Internal Medicine, University of Manitoba, Winnipeg, MB, Canada. [84]Department of Diabetology, APHP, Paris, France. [85]Sorbonne Université, INSERM, NutriOmic team, Paris, France. [86]Department of Nutrition, Dietetics and Food, Monash University, Melbourne, Victoria, Australia. [87]Monash Centre for Health Research and Implementation, Monash University, Clayton, VIC, Australia. [88]Health Management Center, The Second Affiliated Hospital of Chongqing Medical University, Chongqing

Medical University, Chongqing, China. [89]MRC/Wits Developmental Pathways for Health Research Unit, Department of Paediatrics, Faculty of Health Sciences, University of the Witwatersrand, Johannesburg, South Africa. [90]Channing Division of Network Medicine, Brigham and Women's Hospital, Boston, MA, USA. [91]Sydney Brenner Institute for Molecular Bioscience, Faculty of Health Sciences, University of the Witwatersrand, Johannesburg, South Africa. [92]Department of Women and Children's health, King's College London, London, UK. [93]Lifecourse Epidemiology of Adiposity and Diabetes (LEAD) Center, University of Colorado Anschutz Medical Campus, Aurora, CO, USA. [94]Section of Adult and Pediatric Endocrinology, Diabetes and Metabolism, Kovler Diabetes Center, University of Chicago, Chicago, USA. [95]Department of Pediatrics, Riley Hospital for Children, Indiana University School of Medicine, Indianapolis, IN, USA. [96]Richard L. Roudebush VAMC, Indianapolis, IN, USA. [97]Biomedical Research Institute Girona, IdIBGi, Girona, Spain. [98]Diabetes, Endocrinology and Nutrition Unit Girona, University Hospital Dr Josep Trueta, Girona, Spain. [99]Institute of Health System Science, Feinstein Institutes for Medical Research, Northwell Health, Manhasset, NY, USA. [100]University of California at San Francisco, Department of Pediatrics, Diabetes Center, San Francisco, CA, USA. [101]Division of Endocrinology, Diabetes and Metabolism, Cedars-Sinai Medical Center, Los Angeles, CA, USA. [102]Department of Medicine, Cedars-Sinai Medical Center, Los Angeles, CA, USA. [103]Adelaide Medical School, Faculty of Health and Medical Sciences, The University of Adelaide, Adelaide, Australia. [104]Robinson Research Institute, The University of Adelaide, Adelaide, Australia. [105]Department of Public Health and Novo Nordisk Foundation Center for Basic Metabolic Research, Faculty of Health and Medical Sciences, University of Copenhagen, 1014 Copenhagen, Denmark. [106]Division of Endocrinology and Diabetes, Department of Pediatrics, Sanford Children's Hospital, Sioux Falls, SD, USA. [107]University of South Dakota School of Medicine, E Clark St, Vermillion, SD, USA. [108]Department of Biomedical Informatics, University of Colorado Anschutz Medical Campus, Aurora, CO, USA. [109]Department of Epidemiology, Colorado School of Public Health, Aurora, CO, USA. [110]Royal Devon University Healthcare NHS Foundation Trust, Exeter, UK. [111]Oxford Centre for Diabetes, Endocrinology and Metabolism, University of Oxford, Oxford, UK. [112]Division of General Internal Medicine, Massachusetts General Hospital, Boston, MA, USA. [113]UPMC Children's Hospital of Pittsburgh, Pittsburgh, PA, USA. [114]Center for Translational Immunology, Benaroya Research Institute, Seattle, WA, USA. [115]Department of Medicine, Northwestern University Feinberg School of Medicine, Chicago, IL, USA. [116]Department of Pathology & Molecular Medicine, McMaster University, Hamilton, Canada. [117]Population Health Research Institute, Hamilton, Canada. [118]Department of Translational Medicine, Medical Science, Novo Nordisk Foundation, Tuborg Havnevej 19, 2900 Hellerup, Denmark. [119]Department of Diabetes and Endocrinology, Nelson R Mandela School of Medicine, University of KwaZulu-Natal, Durban, South Africa. [120]Center for Public Health Genomics, Department of Public Health Sciences, University of Virginia, Charlottesville, VA, USA. [121]Division of Epidemiology and Community Health, School of Public Health, University of Minnesota, Minneapolis, MN, USA. [122]Department of Chronic Diseases and Metabolism, Clinical and Experimental Endocrinology, KU Leuven, Leuven, Belgium. [123]School of Health and Wellbeing, College of Medical, Veterinary and Life Sciences, University of Glasgow, Glasgow, UK. [124]Department of Obstetrics, Gynecology, and Reproductive Biology, Massachusetts General Hospital and Harvard Medical School, Boston, MA, USA. [125]Sanford Children's Specialty Clinic, Sioux Falls, SD, USA. [126]Department of Pediatrics, Sanford School of Medicine, University of South Dakota, Sioux Falls, SD, USA. [127]Department of Biostatistics, Johns Hopkins Bloomberg School of Public Health, Baltimore, Maryland, USA. [128]Centre for Physical Activity Research, Rigshospitalet, Copenhagen, Denmark. [129]Institute for Sports and Clinical Biomechanics, University of Southern Denmark, Odense, Denmark. [130]Department of Medicine, Division of Endocrinology, Diabetes and Metabolism, Indiana University School of Medicine, Indianapolis, IN, USA. [131]AMAN Hospital, Doha, Qatar. [132]Department of Preventive Medicine, Division of Biostatistics, Northwestern University Feinberg School of Medicine, Chicago, IL, USA. [133]Institute of Molecular and Genomic Medicine, National Health Research Institutes, Taipei City, Taiwan. [134]Divsion of Endocrinology and Metabolism, Taichung Veterans General Hospital, Taichung, Taiwan. [135]Division of Endocrinology and Metabolism, Taipei Veterans General Hospital, Taipei, Taiwan. [136]Center for Interventional Immunology, Benaroya Research Institute, Seattle, WA, USA. [137]Barbara Davis Center for Diabetes, University of Colorado Anschutz Medical Campus, Aurora, CO, USA. [138]University Hospital of Tübingen, Tübingen, Germany. [139]Institute of Diabetes Research and Metabolic Diseases (IDM), Helmholtz Center Munich, Neuherberg, Germany. [140]Steno Diabetes Center Aarhus, Aarhus University Hospital, Aarhus, Denmark. [141]University of Newcastle, Newcastle upon Tyne, UK. [142]Sections on Genetics and Epidemiology, Joslin Diabetes Center, Harvard Medical School, Boston, MA, USA. [143]Department of Clinical Pharmacy and Pharmacology, University Medical Center Groningen, Groningen, The Netherlands. [144]Gastroenterology, Baylor College of Medicine, Houston, TX, USA. [145]Department of Endocrinology, University Hospitals Leuven, Leuven, Belgium. [146]Sorbonne University, Inserm U938, Saint-Antoine Research Centre, Institute of Cardiometabolism and Nutrition, Paris 75012, France. [147]Department of Endocrinology, Diabetology and Reproductive Endocrinology, Assistance Publique-Hôpitaux de Paris, Saint-Antoine University Hospital, National Reference Center for Rare Diseases of Insulin Secretion and Insulin Sensitivity (PRISIS), Paris, France. [148]Royal Melbourne Hospital Department of Diabetes and Endocrinology, Parkville, Vic, Australia. [149]Walter and Eliza Hall Institute, Parkville, Vic, Australia. [150]University of Melbourne Department of Medicine, Parkville, Vic, Australia. [151]Deakin University, Melbourne, Australia. [152]Department of Epidemiology, Madras Diabetes Research Foundation, Chennai, India. [153]Department of Diabetes and Endocrinology, Guy's and St Thomas' Hospitals NHS Foundation Trust, London, UK. [154]School of Agriculture, Food and Wine, University of Adelaide, Adelaide, Australia. [155]Institut Cochin, Inserm U 10116 Paris, France. [156]Pediatric endocrinology and diabetes, Hopital Necker Enfants Malades, APHP Centre, université de Paris, Paris, France. [157]Department of Medicine, University of Maryland School of Medicine, Baltimore, MD, USA. [158]Department of Epidemiology, Geisel School of Medicine at Dartmouth, Hanover, NH, USA. [159]Nephrology, Dialysis and Renal Transplant Unit, IRCCS—Azienda Ospedaliero-Universitaria di Bologna, Alma Mater Studiorum University of Bologna, Bologna, Italy. [160]Global Center for Asian Women's Health, Yong Loo Lin School of Medicine, National University of Singapore, Singapore, Singapore. [161]Department of Obstetrics and Gynecology, Yong Loo Lin School of Medicine, National University of Singapore, Singapore, Singapore. [162]Kaiser Permanente Northern California Division of Research, Oakland, California, USA. [163]Department of Epidemiology and Biostatistics, University of California San Francisco, California, USA. [164]National Institute of Diabetes and Digestive and Kidney Diseases, National Institutes of Health, Bethesda, MD, USA. [165]Department of Health Research Methods, Evidence, and Impact, Faculty of Health Sciences, McMaster University, Hamilton, ON, Canada. [166]Ann & Robert H. Lurie Children's Hospital of Chicago, Department of Pediatrics, Northwestern University Feinberg School of Medicine, Chicago, IL, USA. [167]Department of Clinical and Organizational Development, Chicago, IL, USA. [168]American Diabetes Association, Arlington, Virginia, USA. [169]College of Medicine and Health Sciences, University of Gondar, Gondar, Ethiopia. [170]Global Health Institute, Faculty of Medicine and Health Sciences, University of Antwerp, 2160 Antwerp, Belgium. [171]Department of Medicine and Kovler Diabetes Center, University of Chicago, Chicago, IL, USA. [172]School of Nursing, Faculty of Health Sciences, McMaster University, Hamilton, Canada. [173]Division of Endocrinology, Metabolism, Diabetes, University of Colorado, Boulder, CO, USA. [174]Department of Clinical Medicine, School of Medicine, Trinity College Dublin, Dublin, Ireland. [175]Department of Endocrinology, Wexford General Hospital, Wexford, Ireland. [176]Department of Medicine, Prtizker School of Medicine, University of Chicago, Chicago, IL, USA. [177]Department of Genetics, Stanford School of Medicine, Stanford University, Stanford, CA, USA. [178]Faculty of Health, Aarhus University, Aarhus, Denmark. [179]Departments of Pediatrics and Medicine and Kovler Diabetes Center, University of Chicago, Chicago, USA. [180]Sanford Research, Sioux Falls, SD, USA. [181]University of Washington School of Medicine, Seattle, WA, USA. [182]Department of Population Medicine, Harvard Medical School, Harvard Pilgrim Health Care Institute, Boston, MA, USA. [183]Department of Medicine, Universite de Sherbrooke, Sherbrooke, QC, Canada. [184]Department of Internal Medicine, Seoul National University College of Medicine, Seoul National University Hospital, Seoul, Republic of Korea. [185]Joslin Diabetes

Center, Harvard Medical School, Boston, MA, USA. [186]Charles Bronfman Institute for Personalized Medicine, Icahn School of Medicine at Mount Sinai, New York, NY, USA. [187]Broad Institute, Cambridge, MA, USA. [188]Division of Metabolism, Digestion and Reproduction, Imperial College London, London, UK. [189]Department of Diabetes & Endocrinology, Imperial College Healthcare NHS Trust, London, UK. [190]Department of Diabetology, Madras Diabetes Research Foundation & Dr. Mohan's Diabetes Specialities Centre, Chennai, India. [191]Department of Medicine, Faculty of Medicine and Health Sciences, University of Auckland, Auckland, New Zealand. [192]Auckland Diabetes Centre, Te Whatu Ora Health New Zealand, Auckland, New Zealand. [193]Medical Bariatric Service, Te Whatu Ora Counties, Health New Zealand, Auckland, New Zealand. [194]Oxford NIHR Biomedical Research Centre, University of Oxford, Oxford, UK. [195]University of Cambridge, Metabolic Research Laboratories and MRC Metabolic Diseases Unit, Wellcome-MRC Institute of Metabolic Science, Cambridge, UK. [196]Department of Epidemiology & Public Health, University of Maryland School of Medicine, Baltimore, MD, USA. [197]Department of Internal Medicine, Division of Metabolism, Endocrinology and Diabetes, University of Michigan, Ann Arbor, MI, USA. [198]AdventHealth Translational Research Institute, Orlando, FL, USA. [199]Pennington Biomedical Research Center, Baton Rouge, LA, USA. [200]MRC Human Genetics Unit, Institute of Genetics and Cancer, University of Edinburgh, Edinburgh, UK. [201]Yale School of Medicine, New Haven, CT, USA. [202]Faculty of Medicine and Health, University of Sydney, Sydney, NSW, Australia. [203]Department of Endocrinology, Royal Prince Alfred Hospital, Sydney, NSW, Australia. [204]Kaiser Permanente Northwest, Kaiser Permanente Center for Health Research, Portland, OR, USA. [205]Clinial Research, Steno Diabetes Center Copenhagen, Herlev, Denmark. [206]Department of Clinical Medicine, Faculty of Health and Medical Sciences, University of Copenhagen, Copenhagen, Denmark. [207]Department of Endocrinology and Diabetology, University Hospital Düsseldorf, Heinrich Heine University Düsseldorf, Moorenstr. 5, 40225 Düsseldorf, Germany.

