## [Peer Review File · Communications Medicine]

Reviewers' comments:

Reviewer #1 (Remarks to the Author):

This article provides a comprehensive analysis of monogenic diabetes testing, covering neonatal diabetes, gestational diabetes mellitus (GDM), and non-obese individuals under 30. The authors distill key findings from a large number of publications to offer recommendations.

Key points include the importance of early diagnosis of neonatal diabetes before six months of age using a 23-gene panel. Testing for monogenic diabetes in babies aged 12-24 months yielded no cases. For GDM, testing non-obese women for GCK etiology is recommended, with a yield of 1%-6% in unselected cases and up to 31% in non-obese women.

The article also explores monogenic diabetes testing in non-obese individuals under 30 and discusses triage strategies and biomarkers for differentiating genetic etiologies. The authors highlight the benefits of Next-Generation Sequencing (NGS) over Sanger sequencing, and the importance of gene-disease validity.

This is an important contribution to our ongoing efforts on making monogenic diabetes testing beneficial for patients. There are several points that should be addressed by authors to further improve the quality of this important article.

1. How can we improve the yield of diagnosis of monogenic diabetes cases in babies aged 12-24 months?
2. What strategies could enhance the diagnosis of monogenic diabetes in individuals under 30?
3. How might recent advances in biomarker research improve the detection of monogenic diabetes?
4. What are the limitations of the MODY risk calculator and how can they be addressed in future research?
5. Are there cost-benefit considerations between NGS and Sanger sequencing that could impact their usage in different healthcare systems?
6. How can we further enhance the diagnostic yield of monogenic diabetes beyond the improvements provided by NGS?
7. Could more standardized global guidelines be developed to determine causative genes in monogenic diabetes?
8. How do geographical and cultural factors affect the prevalence and detection of specific monogenic diabetes subtypes?
9. What potential future technologies could be used to improve the testing of monogenic diabetes?

Reviewer #2 (Remarks to the Author):

Murphy et al have undertaken a systematic review to address the following questions:

Question 1: To investigate the evidence for who to test for variants in monogenic diabetes genes

Question 2: How to test for monogenic diabetes

Question 3: What is the basis for considering a gene as a cause of monogenic diabetes?

Question 4: On what basis should a variant be considered a cause of monogenic diabetes?

Question 5: How should a variant in a monogenic diabetes gene be reported?

Question 6: What are the next steps after a diagnosis of monogenic diabetes?

Question 7: What are the current challenges for the field in precision diagnostics for monogenic diabetes?

The authors did not consider the evidence underpinning the link between the genetic test result and clinical management; and prognostics; as these will be covered by separate systematic reviews.

The authors have clearly undertaken a major piece of work that will be a useful contribution to the literature. However, firstly, by covering such a broad spectrum of the diagnostic pathway, they are including areas that have been and/or will be covered by other reviews (for example Q5 and the existing ACMG guidelines; Q6 and the upcoming systematic reviews by this group on the link between genetic test result and clinical management). Secondly, the title and abstract is misleading, giving the impression that this systematic review covers all forms of monogenic diabetes; whereas the focus is very much on what used to be called MODY genes. Thirdly, there are some methodological issues that need further explanation. I have broken down these general comments in the following list:

1. Plain Language Summary. Please get someone to write this who is used to writing plain language summaries. This is not a plain language summary. You might want to use the Flesch easy read score to check your text. Your summary came out with a score 29.5, reading level College Graduate (very difficult to read).

2. The grouping of the results into research questions is very helpful. I suggest list these research questions that you are going to answer, in the introduction.

3. Methods. There needs to be a definition of which monogenic diabetes genes you are investigating in the systematic review. Are you including genes where diabetes is the only phenotype? Or the cardinal feature with other features occurring in only a proportion of people? Or a component of a monogenic disease where diabetes only occurs in a proportion of patients? Diabetes gene panels test for genes in all 3 of these categories; but it was only in the discussion that you mentioned you were not able to provide recommendations on selecting whom to provide genetic testing for syndromic forms including mitochondrial diabetes and deafness, severe insulin resistance, lipodystrophy, or obesity related diabetes phenotypes with monogenic aetiologies. This needs to be made explicit much higher up the paper, preferably in the abstract.

4. Methods. Reference 33 QUADAS guidelines. These are very helpful to assess risk of bias, and in their Table 1, they include a signalling question: 'was a case-control design avoided?'. In the accompanying text, they explain that studies enrolling participants with known disease and a control group without the condition may exaggerate diagnostic accuracy; and excluding patients with 'red flags' for the target condition who may be easier to diagnose may lead to underestimation of diagnostic accuracy'. Has this been misinterpreted in the case of a study you excluded (reference 46 (Riveline et al, Clinical and metabolic features of adult-onset diabetes caused by ABCC8 mutations)? This study screened 7 adult subjects from 3 NDM families, and 139 T2DM patients who were well controlled with sulfonylureas, for ABCC8 mutations. I may be mistaken, but this sounds like a study with an appropriate control group for screening for ABCC8 mutations and therefore should have been included.

5. Methods. Reference 34 is wrong, not the Canadian guidelines on grading evidence for diabetes studies

6. Methods. In relation to the Canadian guidelines, you say that independent interpretation of the diagnostic standard (the reference genetic test result) without knowledge of the triage test result (item 7 of the bias tool) was considered to always be the case. This is not always the case. In practice, interpretation of the genetic test result is often done with knowledge of the clinical features, and sometimes in consultation with a clinical expert or expert panel.

7. Results. Page 20 'or have retained C-peptide (for lowering probability of type 1 diabetes) and those without obesity (for lowering probability of type 2 diabetes)'. Would be helpful to define terms here, definition of retained C-peptide, definition of obesity.

8. Results Q5: how should a variant in a monogenic diabetes gene be reported? I think this question is covered by existing guidelines, so am not sure what this review adds to this question. There is a pertinent issue in relation to Q5 that could be covered: the need to produce genetic testing results reports that are searchable. Searching EPRs to identify patients with specific genomic changes is impossible at scale due to lack of structured, interpreted, searchable data in EPRs (as often results are in a pdf attachment). It would be really helpful to move towards an interoperable system for structured reporting that can incorporate interpreted gene variants into searchable systems that can interrogate genotypic data.

9. Results Q6. What are the next steps after a diagnosis of monogenic diabetes? I feel you are trying to make practical recommendations for providing diagnosis results and clinical follow-up; but this is a specialist area and have you had genetic counsellor input here? As an example, there is no mention of the guilt that parents may feel (It's not your fault). I feel that trying to cover genetic counselling is outside the scope of this manuscript. As another example, there is little mention of assessing the implications of disease gene variants that are associated with an atypical course of disease; or those monogenic diabetes genes where the development of diabetes is heavily influenced by environmental factors. Your review might benefit from omitting Q6 as it is hard to give advice about clinical follow-up without mentioning the need for referral to specialist MDT teams (for instance see ORPHANET directory of specialists/centres); and I think Q6 will be covered by a separate systematic review from this group?

10. Page 33 'In this case, the high risk of developing diabetes or hyperglycaemia should be emphasised and a plan for monitoring blood SUGARS should be developed..'. Please avoid the term 'sugars' and just use 'glucose'.

11. Results Q7: what are the current challenges for the field in precision diagnostics for monogenic diabetes? There are still some gaps in this section. Missing: the increased numbers of patients being diagnosed due to NGS, and diabetes genes on multiple gene panels, for instance obesity, retinal dystrophy, deafness, etc. This raises issues in the capacity of medical services to provide management for these patients. Part of the justification for your review is to identify people who may benefit from precision medicine- so it would be helpful to mention the ongoing referral pathways to deliver precision medicine, and challenges to delivery; even if this will be covered in depth in a future systematic review.

12. Discussion. 'Our review was limited by the availability of relevant studies, and sometimes there were not enough high-quality studies to draw meaningful conclusions. Hence, we were not able to address Questions 3-7 initially or ultimately (for Questions 6-7) by a systematic review using the

method offered by Covidence. However, the co-authors have been working on diagnostics of monogenic diabetes for 10-30 years and are experts in the field. We therefore used expert opinion for the Questions 3-7'. In that case, for questions 3-7 this is not a systematic review, but a consensus review. It needs to be made explicit in the abstract what is being addressed by a systematic review, and what by consensus expert opinion.

Referee expertise:

Referee #1: personalised medicine and monogenic diabetes diagnostics (clinical)

Referee #2: monogenic diabetes diagnostics and systematic reviews (clinical)

We thank both reviewers for their constructive comments on our review and appreciate the time it will have taken them to carefully evaluate our manuscript. Below we address their individual comments and highlight the changes we have made to our review.

Reviewers' comments:**Reviewer #1 (Remarks to the Author):**

This article provides a comprehensive analysis of monogenic diabetes testing, covering neonatal diabetes, gestational diabetes mellitus (GDM), and non-obese individuals under 30. The authors distill key findings from a large number of publications to offer recommendations.

Key points include the importance of early diagnosis of neonatal diabetes before six months of age using a 23-gene panel. Testing for monogenic diabetes in babies aged 12-24 months yielded no cases. For GDM, testing non-obese women for GCK etiology is recommended, with a yield of 1%-6% in unselected cases and up to 31% in non-obese women.

The article also explores monogenic diabetes testing in non-obese individuals under 30 and discusses triage strategies and biomarkers for differentiating genetic etiologies. The authors highlight the benefits of Next-Generation Sequencing (NGS) over Sanger sequencing, and the importance of gene-disease validity.

This is an important contribution to our ongoing efforts on making monogenic diabetes testing beneficial for patients. There are several points that should be addressed by authors to further improve the quality of this important article.

We appreciate the reviewers' suggestions for additional areas which could be discussed and where possible, given limitations on space, we have expanded our discussion to include them.

1. How can we improve the yield of diagnosis of monogenic diabetes cases in babies aged 12-24 months?

The highest yield of detected monogenic neonatal diabetes cases versus all tested cases of diabetes occurring below the age of 12 months is likely due to the nature of most monogenic neonatal diabetes presenting early in life and decreasing in prevalence with age of onset beyond 12 months as per the first paragraph of the discussion. Further, improvements in our ability to diagnose type 1 diabetes using genetic risk scores and autoantibodies will also help differentiate those with monogenic diabetes. The following text has been added on page 37, to this part of the discussion.

“The ongoing detection of the commonest subtype of neonatal diabetes due to KCNJ11/ABCC8 mutations who can be transferred to inexpensive sulfonylurea treatment makes testing all those below 12 months still cost-saving, so long as 3% of those screened have a monogenic diabetes diagnosis that is treatment-changing⁸. Further large-scale studies need to be performed using large-gene panels in those with diabetes diagnosed between 12 and 24 months to ascertain further triaging features that could improve the yield of diagnosis of monogenic diabetes cases amongst the increasing majority who would have type 1 diabetes during early childhood, such as with type 1 diabetes related antibodies and type 1 diabetes genetic risk score. The prevalence of type 2 diabetes is generally only seen with severe obesity in early childhood with onset generally well above 24 months of age, so excluding obesity in those selected for genetic testing is less of a consideration for this young age group.”

2. What strategies could enhance the diagnosis of monogenic diabetes in individuals under 30?

We thank the reviewer for this question. The detection of GCK etiology, may be enhanced with the consideration of some additional biochemical features and more nuanced selection by normal weight rather than non-obese. We have included additional glucose and BMI data in table 1 and supplementary table 3, and revised recommendations 2 and 3 in box 1 as follows (on page 75 of manuscript) and text on page 18-19:

Recommendation 2 “Women with gestational diabetes without obesity should be tested for GCK etiology” is now revised to “Women with gestational diabetes and fasting glucose above 5.5mmol/L without obesity should be tested for GCK etiology”*

Recommendation 3. “Those with persisting, mild hyperglycemia at any age in absence of obesity should be tested for GCK etiology” revised to “Those with persisting, mild hyperglycemia (HbA1c 38-62, or fasting glucose 5.5-7.8mmol/L) at any age, in absence of obesity should be tested for GCK etiology”*

*Asterisked footnote for Box 1 *By selecting those who are of normal weight or underweight (rather than those who are non-obese) to offer genetic testing to, increases the specificity but reduces the sensitivity for detecting GCK and may be considered a less costly approach.*

Text page 18-19: The use of fasting glucose of 5.5mmol/L or higher was demonstrated to have the highest yield (3/118) for GCK, (only 1/129 women with fasting glucose below 5.5mmol/L found to have GCK etiology and 0/109 women with fasting glucose below 5.⁵⁰. In this 2014 study, the tradeoff between specificity and sensitivity of detection for GCK among the women with GDM (and fasting glucose above 5.5mmol/L), by various BMI thresholds is shown (table 1).

In making these corrections, we spotted that the yield for GCK from ref 53 was incorrectly stated as 31% in non-obese women with GDM. This was the highest yield in this category of papers evaluated so makes the highest level of evidence level 2 and the arising recommendation now grade B. This has been corrected on page 18 (text), page 56 (table 1), page 75 (box text 1), and supplementary table 3.

In improving the correct classification of common types of diabetes with use of antibodies or genetic risk score and including such a systematic approach to phenotyping diabetes into clinical pathways will enhance accuracy of diagnosis and detection of monogenic diabetes. However, this comes at a higher cost and may not be affordable in many healthcare contexts. The following text has been added to the discussion to capture this issue on page 38-39.

The recommendations for restricting testing to the selected groups of individuals with diabetes provided in this systematic review, have considered the highest diagnostic yields reported in the literature, balancing high specificity of clinical and/or biochemical features with high sensitivity for MD. As sensitivity for identifying MD subtypes increases with the use of less stringent clinical characteristics, the number needed to test to find one positive case increases, so choosing absolute thresholds of key characteristics is a matter of balancing the costs of the genetic test, against the benefits of correctly identifying individuals with MD. Different health systems should consider adapting these recommended thresholds for their population contexts to allow for systematic genetic testing in those with diabetes whose probability of having a MD etiology meets their cost-effectiveness threshold for implementing through clinical care pathways for diabetes.

Currently, we cannot afford to offer a genetic test to everyone with diabetes. Such an “all testing” approach would find every case of monogenic diabetes but would have a low diagnostic yield. In a USA study, MD genetic screening approach was modelled as being cost-effective at 6% yield and cost-saving at 30% yield for GCK, HNF1A, HNF4A, with an estimated combined prevalence of 2%, implicating the most clinically actionable changes to therapy.¹²² In a UK study, selecting individuals for genetic testing among those diagnosed below 30 years, using either a clinical prediction model or a biomarker strategy (with negative antibodies and retained C-peptide) was deemed cost-saving, assuming cost-benefit from stopping insulin treatment for misdiagnosed type 1 diabetes in this age group.¹²³ This model assumed a prevalence of the most cost-saving MD types as 2.4% (GCK 0.7%, HNF1A 1.5% and HNF4A 0.2%). Clearly, once MD has been identified in a proband, cascade screening of family members maximizes cost-savings, given the yield is at least 50% in autosomal dominant subtypes of MD.

3. How might recent advances in biomarker research improve the detection of monogenic diabetes?

The use of specific biomarkers is discussed on p20-21 of the manuscript and in Table 2. A specific biomarker for one of the commoner forms of monogenic diabetes could be used to guide individual gene sequencing. An ideal biomarker would be cheap, easily measured, not depend on ethnicity and be unaffected by clinical confounders such as obesity. In reality HNF1A diabetes or GCK-related hyperglycaemia are the only subtypes common enough to be a target for such an approach. Several extra-pancreatic HNF1A-biomarkers have been investigated both alone and in combination with other features (detailed in Table 2), and while not meeting all the ideal criteria in some studies do appear to reduce the number of individuals requiring investigation for HNF1A diabetes while in others did not offer any advantage. This approach would also require both Sanger sequencing for the individual genes and the NGS panel to be available in the lab.

We have edited the following text on page 22 to reflect the above:

“Other non-routine biomarkers such as lipid fractions and glycan moieties regulated by HNF1A have been explored for distinguishing HNF1A-diabetes from other diabetes subtypes, but these have added complexity and cost above clinical features without better informing whom to test for the greater yield of monogenic diabetes provided from large-gene panel testing”.

4. What are the limitations of the MODY risk calculator and how can they be addressed in future research?

The MODY calculator currently has limitations in the age of diagnosis (35yrs) and ethnicity (only validated in Europeans). Risks of GCK and HNF1A/HNF4A etiology are calculated together, whereas there are differences between characteristics of GCK and HNF1A/HNF4A subtypes of MD. Biomarkers such as antibodies, c-peptide and hsCRP are not included in the model, although it is suggested to use the Type 1 markers alongside. The calculator also compares the characteristics to a group of already diagnosed MD cases which is likely to bias towards those meeting classic criteria. Limited numbers of younger Type 2 cases were used in the model development. Clearly the addition of data from other ethnic groups would be the most useful addition and we understand this is planned.

We have edited the following text on p22 to reflect the main limitations.

“However, limitations of this calculator include lack of validation for non-European population, those diagnosed with diabetes above 35 years, those with other forms of monogenic diabetes, and weaker performance in insulin-treated patients where the probability of type 1 diabetes is high.”

5. Are there cost-benefit considerations between NGS and Sanger sequencing that could impact their usage in different healthcare systems?

We thank the reviewer for this comment. As stated on page 24, the high diagnostic yield when performing Sanger sequencing of GCK in patients with a clinical suspicion of GCK or Sanger sequencing for detecting the common and most clinical impactful neonatal diabetes monogenic subtypes of KCNJ11, ABCC8, mean these could be used in some healthcare systems under cost-benefit considerations.

6. How can we further enhance the diagnostic yield of monogenic diabetes beyond the improvements provided by NGS?

We thank the reviewer for their suggestion however we feel we have already commented on this (see below) in our discussion.

“What is needed next? Our systematic review reveals that improved access to genetic testing for monogenic diabetes to prevent health disparities is important. There are issues regarding equity and utility in non-European countries where background prevalence of type 2 diabetes is higher. Moreover, type 1 diabetes genetic risk score (a tool for using common susceptibility variants for type 1 diabetes to pre-assess the likelihood of having type 1 vs. other types of diabetes) data has not been well characterized in these countries. Another step is generation of and access to systematic measurements of autoantibodies and C-peptide for people diagnosed with diabetes under the age of 45 years with the addition of validated ancestry-appropriate type 1 diabetes genetic risk score data. This information would be advantageous to better discriminate monogenic diabetes from type 1 diabetes. What is more, improvement in de-identified case-sharing platforms is needed to promote maximizing the ability to gather the evidence needed to evaluate pathogenicity. As such, continued work by expert panels such as the MDEP VCEP is warranted to develop guidelines for which gene variants should be considered causative of monogenic diabetes as well as applications of the guidelines tailored to additional monogenic diabetes types and genes. One relevant instrument is generation of deep mutational scanning maps of monogenic diabetes genes to aid variant classification. It is also important to remember that the genetic and genomic testing landscape is ever evolving, with a strong possibility of universal genome sequencing in on having adequate tools, expertise, and workforce for interpretation.”

7. Could more standardized global guidelines be developed to determine causative genes in monogenic diabetes?

There are already standardized guidelines through the ClinGen monogenic diabetes expert review panels which have international representation.

8. How do geographical and cultural factors affect the prevalence and detection of specific monogenic diabetes subtypes?

The higher background prevalence of type 2 diabetes in non-European populations is highlighted in the discussion on page 41 to which the following has been added:

“Ethnicity dependent thresholds for overweight or obese categories need to be considered when implementing the recommendations of whom to test in different geographical regions with distinct ethnic groups. Cultural factors may also influence the acceptability of screening for MD which needs further study and education.”

9. What potential future technologies could be used to improve the testing of monogenic diabetes?

Although a fascinating area we feel this comment is beyond the scope of our review.

Reviewer #2 (Remarks to the Author):

Murphy et al have undertaken a systematic review to address the following questions:

Question 1: To investigate the evidence for who to test for variants in monogenic diabetes genes

Question 2: How to test for monogenic diabetes

Question 3: What is the basis for considering a gene as a cause of monogenic diabetes?

Question 4: On what basis should a variant be considered a cause of monogenic diabetes?

Question 5: How should a variant in a monogenic diabetes gene be reported?

Question 6: What are the next steps after a diagnosis of monogenic diabetes?

Question 7: What are the current challenges for the field in precision diagnostics for monogenic diabetes?

The authors did not consider the evidence underpinning the link between the genetic test result and clinical management; and prognostics; as these will be covered by separate systematic reviews.

The authors have clearly undertaken a major piece of work that will be a useful contribution to the literature. However, firstly, by covering such a broad spectrum of the diagnostic pathway, they are including areas that have been and/or will be covered by other reviews (for example Q5 and the existing ACMG guidelines; Q6 and the upcoming systematic reviews by this group on the link between genetic test result and clinical management). Secondly, the title and abstract is misleading, giving the impression that this systematic review covers all forms of monogenic diabetes; whereas the focus is very much on what used to be called MODY genes. Thirdly, there are some methodological issues that need further explanation. I have broken down these general comments in the following list:

We thank the reviewer for their constructive comments on our review. We have indeed selected to cover a broad spectrum of the diagnostic pathway as our intention was to provide the best evidence in a single article. The systematic review is designed to cover all forms of monogenic diabetes that are able to be detected using various gene sequencing technologies in a minimum of 100 screened individuals with diabetes, who are not already known to have a monogenic etiology so we feel that the title and abstract fit with this.

1. Plain Language Summary. Please get someone to write this who is used to writing plain language summaries. This is not a plain language summary. You might want to use the Flesch easy read score to check your text. Your summary came out with a score 29.5, reading level College Graduate (very difficult to read).

We thank the reviewer for this helpful feedback and have revised our plain language summary on page 2.

Some diabetes, called monogenic diabetes, is caused by change in a gene. It is important to know if you have this kind of diabetes because treatment can differ. Some treatments work better, and some people can change from insulin shots to tablets. Relatives can also be offered a test. Genetic testing is needed to diagnose monogenic diabetes but is expensive. It is not possible to test every person with diabetes for monogenic diabetes. We looked at research by doctors and scientists on who should be tested and what test to use. This article will provide recommendations for doctors and health care providers.

2. The grouping of the results into research questions is very helpful. I suggest list these research questions that you are going to answer, in the introduction.

We thank the reviewer for this helpful feedback. We note that we have already included the following in our introduction.

“To investigate the evidence for who to test for monogenic diabetes, how to test them and how to interpret a gene variant, we set out to systematically review the yield of monogenic diabetes using different criteria to select people with diabetes for genetic testing and the technologies used. In addition, we sought to evaluate current guidelines for genetic testing for monogenic diabetes using a systematic review and grading of the studies available”.

Since this already sets up our questions and due to limitations on space, we have selected not to do add this to the introduction.

3. Methods. There needs to be a definition of which monogenic diabetes genes you are investigating in the systematic review. Are you including genes where diabetes is the only phenotype? Or the cardinal feature with other features occurring in only a proportion of people? Or a component of a monogenic disease where diabetes only occurs in a proportion of patients? Diabetes gene panels test for genes in all 3 of these categories; but it was only in the discussion that you mentioned you were not able to provide recommendations on selecting whom to provide genetic testing for syndromic forms including mitochondrial diabetes and deafness, severe insulin resistance, lipodystrophy, or obesity related diabetes phenotypes with monogenic aetiologies. This needs to be made explicit much higher up the paper, preferably in the abstract.

We included all known monogenic diabetes genes in which the presenting phenotype was diabetes or mild hyperglycemia in whom the yield of monogenic diabetes was provided. The search strategy is listed in the methods section on page 9 and refers to the genes of interest as specified in the Supplementary table 1. While the syndromic forms including MIDD, SIRS, lipodystrophy were included in the search strategy, there were very few papers that included at least 100 tested individuals (listed under inclusion/exclusion criteria on page 10) to permit recommendations on selecting whom to provide genetic testing for when these additional features were present. We have edited the limitation sentence in the discussion on page 40 to clarify this.

“Whilst syndromic forms of monogenic diabetes (such as mitochondrial diabetes and deafness, severe insulin resistance, lipodystrophy) were included in the search strategy, there were not sufficient papers that included at least 100 genetically tested individuals to permit graded recommendations on whom to select for genetic testing when these additional features were present.”

4. Methods. Reference 33 QUADAS guidelines. These are very helpful to assess risk of bias, and in their Table 1, they include a signalling question: 'was a case-control design avoided?'. In the accompanying text, they explain that studies enrolling participants with known disease and a control group without the condition may exaggerate diagnostic accuracy; and excluding patients with 'red flags' for the target condition who may be easier to diagnose may lead to underestimation of diagnostic accuracy'. Has this been misinterpreted in the case of a study you excluded (reference 46 (Riveline et al, Clinical and

metabolic features of adult-onset diabetes caused by ABCC8 mutations)? This study screened 7 adult subjects from 3 NDM families, and 139 T2DM patients who were well controlled with sulfonylureas, for ABCC8 mutations. I may be mistaken, but this sounds like a study with an appropriate control group for screening for ABCC8 mutations and therefore should have been included.

Thank you for this comment. We can confirm that the paper by Riveline et al was included in our review for Question 1 (reference 46) and is listed in Supplementary Table 3

5. Methods. Reference 34 is wrong, not the Canadian guidelines on grading evidence for diabetes studies

We thank the reviewer for spotting this error. We have updated the reference in the main text. The correct citation is below.

Methods. Diabetes Canada Clinical Practice Guidelines Expert Committee; Sherifali D, Rabi DM, McDonald CG, Butalia S, Campbell DJT, Hunt D, Leung AA, Mahon J, McBrien KA, Palda VA, Banfield L, Sanger S, Houlden RL.

Can J Diabetes. 2018 Apr;42 Suppl 1:S6-S9. doi: 10.1016/j.jcjd.2017.10.002.

PMID: 29650113 No abstract available.

6. Methods. In relation to the Canadian guidelines, you say that independent interpretation of the diagnostic standard (the reference genetic test result) without knowledge of the triage test result (item 7 of the bias tool) was considered to always be the case. This is not always the case. In practice, interpretation of the genetic test result is often done with knowledge of the clinical features, and sometimes in consultation with a clinical expert or expert panel.

Item 7 was uniformly graded as true, because those instances in which the interpretation of the genetic test was done with more detailed knowledge of the clinical features could not be gleaned from the papers reviewed. In either case of being true or false or graded uniformly as N/A, this was not a particularly informative criterion which is already stated in the methods section on page 13 as below.

“Whilst gene variant curation often relies on knowledge of the clinical features and laboratory biomarkers, this criterion was not deemed sufficiently informative for decisions about grading the evidence for the question of whom to offer genetic testing for monogenic diabetes.”

This has also been clarified in Supplementary figure 2 middle box “Criteria for assigning levels of evidence to the published studies of diagnosis”. We noted there was a missing item for assigning level 1 evidence which has been added as “(d) reproducible description of the test and diagnostic standard- items 5 and 6 from JBI checklist”. We have noted in this Supplementary figure 2 middle box that criteria (a) independent interpretation of triage test result without knowledge of the result of the diagnostic genetic test and (b) independent interpretation of the diagnostic genet test without knowledge of the triage test result, were both assumed to be uniformly positive.

7. Results. Page 20 'or have retained C-peptide (for lowering probability of type 1 diabetes) and those without obesity (for lowering probability of type 2 diabetes)'. Would be helpful to define terms here, definition of retained C-peptide, definition of obesity.

We have added to the discussion on page 41 that the recommendation for testing needs to be adapted to various geographical contexts with different populations of different ethnicities for whom the definitions of obesity will vary. Similarly, definition of retained C-peptide will vary with laboratory assays used hence these were not extracted from each paper systematically.

“Ethnicity dependent thresholds for overweight or obese categories need to be considered when implementing the recommendations of whom to test in different geographical regions with distinct ethnic groups.”

8. Results Q5: how should a variant in a monogenic diabetes gene be reported? I think this question is covered by existing guidelines, so am not sure what this review adds to this question. There is a pertinent issue in relation to Q5 that could be covered: the need to produce genetic testing results reports that are searchable. Searching EPRs to identify patients with specific genomic changes is impossible at scale due to lack of structured, interpreted, searchable data in EPRs (as often results are in a pdf attachment). It would be really helpful to move towards an interoperable system for structured reporting that can incorporate interpreted gene variants into searchable systems that can interrogate genotypic data.

The purpose of including this section was twofold: 1) To summarize the noted existing guidelines on genetic test reporting to an audience that may not be familiar with genetic testing, and 2) to provide specific guidance on how to most effectively apply these standards to the specific case of monogenic diabetes in a way that results in reports that are best positioned to assist clinicians in carrying out optimal management.

9. Results Q6. What are the next steps after a diagnosis of monogenic diabetes? I feel you are trying to make practical recommendations for providing diagnosis results and clinical follow-up; but this is a specialist area and have you had genetic counsellor input here? As an example, there is no mention of the guilt that parents may feel (It's not your fault). I feel that trying to cover genetic counselling is outside the scope of this manuscript. As another example, there is little mention of assessing the implications of disease gene variants that are associated with an atypical course of disease; or those monogenic diabetes genes where the development of diabetes is heavily influenced by environmental factors. Your review might benefit from omitting Q6 as it is hard to give advice about clinical follow-up without mentioning the need for referral to specialist MDT teams (for instance see ORPHANET directory of specialists/centres); and I think Q6 will be covered by a separate systematic review from this group?

We thank the reviewer for their comment. Regarding genetic counsellor input, this section was written by two genetic counsellors with board certification by the American Board of Genetic Counselling, Toni Pollin and Kristin Maloney. The third co-author on this section was Liana Billings, who while not a genetic counsellor, is a board-certified endocrinologist with expertise in monogenic diabetes. Ms. Maloney and Dr. Billings regularly provide genetic counselling including working through genetic testing for patients with suspected monogenic diabetes and are among a very small number of clinicians specializing in this area. Ms. Maloney and Dr. Pollin are the lead and senior authors of a practice resource of the National Society of Genetic Counsellors Practice Guidelines. As this document was accepted for publication after the submission of the current manuscript, it is now in-press and we have added a citation to the current manuscript. In both manuscripts, we note that it is ideal to have a genetic counsellor on the health care team, while also recognizing that there is a need to disseminate knowledge about these issues more widely to health care providers if we are going to meet the needs of the patient population.

10. Page 33 'In this case, the high risk of developing diabetes or hyperglycaemia should be emphasised and a plan for monitoring blood SUGARS should be developed..'. Please avoid the term 'sugars' and just use 'glucose'.

Thank you for spotting this it has been corrected.

11. Results Q7: what are the current challenges for the field in precision diagnostics for monogenic diabetes? There are still some gaps in this section. Missing: the increased numbers of patients being diagnosed due to NGS, and diabetes genes on multiple gene panels, for instance obesity, retinal dystrophy, deafness, etc. This raises issues in the capacity of medical services to provide management for these patients. Part of the justification for your review is to identify people who may benefit from precision medicine- so it would be helpful to mention the ongoing referral pathways to deliver precision medicine, and challenges to delivery; even if this will be covered in depth in a future systematic review.

We thank the reviewer for their comment. We performed a schematic review of the challenges (as outlined in the methods) and focused on the areas that were identified through our review. We have added a comment to the discussion on page 42, to add these are further areas/gaps which could be explored.

“With increased numbers of people being given a genetic diagnosis for their monogenic diabetes It will also be important for future efforts to evaluate the capacity of medical services to provide management for people with a genetic diagnosis of monogenic diabetes.”

12. Discussion. 'Our review was limited by the availability of relevant studies, and sometimes there were not enough high-quality studies to draw meaningful conclusions. Hence, we were not able to address Questions 3-7 initially or ultimately (for Questions 6-7) by a systematic review using the method offered by Covidence. However, the co-authors have been working on diagnostics of monogenic diabetes for 10-30 years and are experts in the field. We therefore used expert opinion for the Questions 3-7'. In that case, for questions 3-7 this is not a systematic review, but a consensus review. It needs to be made explicit in the abstract what is being addressed by a systematic review, and what by consensus expert opinion.

We thank the reviewer for their comment. We note that our abstract already reflects this.

“In this review, we perform a systematic evaluation of the evidence for the clinical and biochemical criteria used to guide selection of individuals with diabetes for genetic testing and review the evidence for the optimal methods for variant detection in genes involved in monogenic diabetes. In parallel we revisit the current ACMG guidelines for genetic testing for monogenic diabetes and provide expert opinion on the interpretation and reporting of genetic tests. We provide a series of recommendations for the field informed by our systematic review, synthesizing evidence, and expert opinion. “

REVIEWERS' COMMENTS:

Reviewer #1 (Remarks to the Author):

Upon revisiting the manuscript, I observe that the authors have thoroughly addressed all the concerns and points I raised in my initial review. The revisions made have enhanced the quality and clarity of the article. I believe the article is now in a polished state and recommend it for publication in its current form.

Reviewer #2 (Remarks to the Author):

Thankyou for your helpful rebuttal letter, and most of my questions have been answered.

General: Manuscript title: I still don't think the manuscript title reflects the manuscript content. You say in your rebuttal that the systematic review is designed to cover all forms of monogenic diabetes, WHO ARE NOT ALREADY KNOWN TO HAVE A MONOGENIC ETIOLOGY. .This should be reflected in the title. In the abstract, you say you perform a systematic review, but in reality this is only for the first 2 of 7 questions you set out to answer; you provide expert opinion on others such as the interpretation and reporting of genetic tests.

Current title: A systematic review of the use of precision diagnostics in monogenic diabetes

More accurate and not misleading: A systematic review and expert opinion on the use of precision diagnostics in monogenic diabetes of unkown ethology.

Rebuttal point 3 Methods. You have clarified in your rebuttal that you included all known monogenic diabetes genes in which the PRESENTING PHENOTYPE was diabetes or mild hyperglycaemia. Please include this clarification (presenting phenotype) in either the search strategy (P9) or inclusion criteria (P10).

Rebuttal point 4 Methods. In the revised manuscript version the paper by Riveline et al is now reference 47 (not 46). You still state (bottom of P17) that the risk criterion for patient selection was high because a case-control study had not been avoided in Riveline's study. Is that correct, as the controls (adults with T2DM well controlled with sulfonylureas) had the same condition (diabetes) as the cases, as opposed to being a control group without the disease?

Rebuttal point 6 Methods. It your manuscript (Item 7 of the bias tool) you state that independent interpretation of the diagnostic standard without knowledge of the triage test result was considered always to be the case.

In your rebuttal you state that Item 7 of the Canadian guidelines bias tool was uniformly graded as true, because those instances in which the interpretation of the genetic test was done with more detailed knowledge of the clinical features could not be gleaned from the papers reviewed. For clarity you should state that that is the reason why you uniformly graded that item as true.

General: Manuscript title: I still don't think the manuscript title reflects the manuscript content. You say in your rebuttal that the systematic review is designed to cover all forms of monogenic diabetes, WHO ARE NOT ALREADY KNOWN TO HAVE A MONOGENIC ETIOLOGY. .This should be reflected in the title. In the abstract, you say you perform a systematic review, but in reality this is only for the first 2 of 7 questions you set out to answer; you provide expert opinion on others such as the interpretation and reporting of genetic tests.

Current title: A systematic review of the use of precision diagnostics in monogenic diabetes

More accurate and not misleading: A systematic review and expert opinion on the use of precision diagnostics in monogenic diabetes of unknown etiology.

Thank you for this suggestion, we have revised the title to: "A systematic review and expert opinion on the use of precision diagnostics for monogenic diabetes."

Rebuttal point 3 Methods. You have clarified in your rebuttal that you included all known monogenic diabetes genes in which the PRESENTING PHENOTYPE was diabetes or mild hyperglycaemia. Please include this clarification (presenting phenotype) in either the search strategy (P9) or inclusion criteria (P10).

The presenting phenotype of "diabetes or mild hyperglycemia" is already included in the inclusion criteria paragraph on page 10, line 3.

Rebuttal point 4 Methods. In the revised manuscript version the paper by Riveline et al is now reference 47 (not 46). You still state (bottom of P17) that the risk criterion for patient selection was high because a case-control study had not been avoided in Riveline's study. Is that correct, as the controls (adults with T2DM well controlled with sulfonylureas) had the same condition (diabetes) as the cases, as opposed to being a control group without the disease?

We have reviewed the Riveline et al study again and apologise for missing this earlier. The reviewer is indeed correct. We have updated the text on page 17 to reflect this. We have also changed the yield from 4/139 to 2/139 in supplementary table 3, given the variant curation of likely causal was provided for 2/139.

Rebuttal point 6 Methods. In your manuscript (Item 7 of the bias tool) you state that independent interpretation of the diagnostic standard without knowledge of the triage test result was considered always to be the case.

In your rebuttal you state that Item 7 of the Canadian guidelines bias tool was uniformly graded as true, because those instances in which the interpretation of the genetic test was done with more detailed knowledge of the clinical features could not be gleaned from the papers reviewed. For clarity you should state that that is the reason why you uniformly graded that item as true.

Thank you for this comment – we have now included this clarification on page 13 of the methods.